# CHESS: Chebyshev Spectral Synthesis For Trajectory Condensation

**Ruituo Wu** [1]  **Hongyu Zhang** [2]  **Qiang Wang** [1]  **Jiawei Du** [3]  **Wei Cui** [4]  **Ce Zhu** [1]  **Bing Li** [1]

## Abstract

Learning from continuous-time trajectories requires modeling multivariate sensor measurements generated by underlying physical or dynamical processes. Under extreme data compression and heterogeneous sampling, directly optimizing synthetic signals as discrete sample values becomes fundamentally misaligned with the underlying *continuous-time physical processes*, often producing high-frequency, non-physical artifacts that overfit specific models and break reuse across architectures and sampling rates. We propose CHESS, a *function-first* synthesis framework shifts optimization from discrete samples to underlying continuous-time signal trajectories. CHESS injects physics-induced structure by jointly enforcing low-rank spatial coherence and piecewise Chebyshev polynomial temporal parameterization, constraining synthesis to a physically meaningful function manifold. We provide theoretical analysis establishing explicit smoothness and stability guaranties. Experiments on diverse sensor testbeds under the dataset distillation protocol demonstrate CHESS consistently outperform state-of-the-art methods with a compression ratios up to $133\times$ for each synthetic sample. Furthermore, CHESS exhibits strong cross-architecture generalization and enables zero-shot adaptation across different sampling resolutions. Code is available at https://github.com/wrtnew/CHESS.

[1]School of Information and Communication Engineering, University of Electronic Science and Technology of China (UESTC), Chengdu, China [2]Shanghai Key Laboratory of Navigation and Location-based Serviced, School of Automation and Intelligent Sensing, Shanghai Jiao Tong University, Shanghai, China [3] Centre for Frontier AI Research (CFAR) & Institute of High Performance Computing (IHPC), Agency for Science, Technology and Research (A*STAR), Singapore [4]Institute for Infocomm Research ($I^2$R), Agency for Science, Technology and Research (A*STAR), Singapore. Correspondence to: Bing Li <bing_li@uestc.edu.cn>.

*Proceedings of the $43^{rd}$ International Conference on Machine Learning*, Seoul, South Korea. PMLR 306, 2026. Copyright 2026 by the author(s).

## 1. Introduction

Continuous trajectories serve as a core abstraction in modeling physical and cyber-physical systems, where information is encoded within the evolving patterns of multivariate data produced by underlying dynamics[1]. These trajectories are ubiquitous across sensing modalities, including wireless signals (Wang et al., 2015; 2017; Li et al., 2021), wearable and inertial sensors (Bhat et al., 2018; Khalifa et al., 2017; Zhang et al., 2022), biomedical measurements (Rubanova et al., 2019), and capture system dynamics beyond what static or frame-based representations can express.

Despite their importance, learning from continuous trajectories remains challenging under practical resource constraints. Modern deep learning models typically rely on large-scale, densely sampled datasets to generalize well (Zhang et al., 2022). The reliance on massive datasets is often infeasible in real-world sensing pipelines, due to the high cost of data collection, storage, and transmission, particularly on edge and embedded platforms. Beyond data volume, a more important challenge lies in the heterogeneity of real-world sensing pipelines: the same physical process may be observed under different sampling rates, sensor configurations, and downstream model architectures, further complicating data reuse.

To support practical deployment, *trajectory condensation* is expected to produce a compact representation of continuous trajectories that remains *reusable* across heterogeneous settings. Concretely, effective trajectory condensation should satisfy three objectives: *i*) *compactness*, enabling extreme data reduction while preserving task-relevant information; *ii*) *cross-architecture reusability*, enabling the compressed representation to generalize across different downstream models; and *iii*) *resolution-agnostic generalization*, supporting reuse under unseen temporal sampling rates. These objectives implicitly favor *representations by structured functions*, with low intrinsic complexity, decoupled from specific architectures and sampling grids.

In contrast, most existing dataset condensation strategies (Yu et al., 2023), e.g., coreset selection (Har-Peled & Mazumdar, 2004), prototype learning (Snell et al., 2017),

[1]In this paper, a *trajectory* is defined as a continuous-time function, whereas a *signal* refers to its discretely sampled counterpart.

and dataset distillation (DD) (Wang et al., 2018), while differing in form, rely on *sample-wise parameterization*, treating both real and synthesized data as discrete samples. This assumption is well aligned with static modalities such as images, where pixel-level representations are natural modeling units. However, when applied to data generated by continuous physical processes, sample-wise parameterization mismatches the underlying data-generating mechanism and fails to satisfy the objectives of trajectory condensation.

Empirical observations in Fig. 1 reveal that the core issue lies in the *ill-posedness* caused by optimizing synthetic trajectories directly within a discrete sample space. By doing so, the condensation process makes a drift from a latent yet *structured function space*, where underlying physical processes govern trajectory generation, to an arbitrary high-dimensional *Euclidean space*, in which the structured constraints are absent. An example of this mismatch is shown in Fig. 1: the synthesized data learns an artifact-dominated pattern, producing high-frequency, uncorrelated spikes, regardless of the strong correlation and temporal smoothness among physical channels. This ill-posedness arises from three closely related aspects: temporally, unconstrained inter-sample behavior violates physical continuity, producing degenerate solutions containing high-frequency, non-physical oscillations; spatially, independent channel modeling ignores intrinsic cross-channel coupling, inducing architecture-specific artifacts; and with respect to sampling, binding representations to fixed grids prevents reuse under unseen resolutions.

To address this challenge, we propose a fundamental shift in paradigm: instead of optimizing *discrete points*, we model *continuous physical trajectories* themselves. We introduce **CHESS** (**CHE**byshev **S**pectral **S**ynthesis), a "function-first" trajectory condensation framework that anchors the synthesis process within physically meaningful manifold. CHESS operates through three tightly coupled components: *i) Low-rank spatial modeling* captures intrinsic cross-channel coupling by restricting multivariate trajectories to a shared low-dimensional subspace, preventing artifacts that emerge from modeling channels in isolation. *ii) Piecewise Chebyshev temporal fitting* parameterizes trajectories using continuous polynomial bases. Inspired by the *Stone-Weierstrass theorem* (Rudin, 1976), this imposes explicit smoothness, suppressing high-frequency "spike" solutions while preserving expressive dynamics. *iii) Continuous trajectory resampling* decouples the representation from any fixed sampling grid. By analytical resampling at arbitrary resolutions, CHESS allows for seamless reuse across heterogeneous sensing configurations without redistillation. Beyond empirical effectiveness, theoretical guaranties ensure CHESS's stability and robustness. Leveraging piecewise Chebyshev polynomial parameterization, we establish explicit bounds on temporal derivatives within each segment, providing

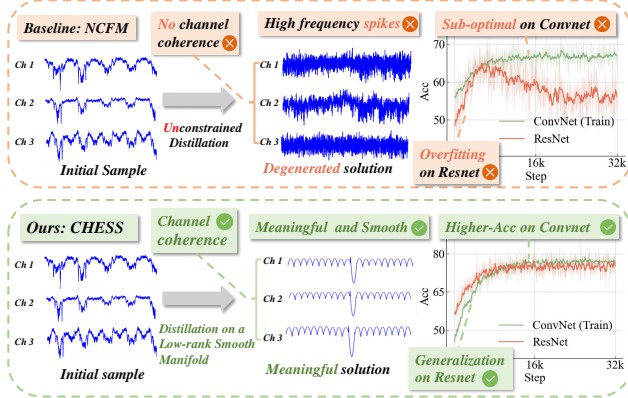

*Figure 1.* Failure modes of vision-based distillation on the MeR dataset (SPC=5, channels Ch1–3). The representative SOTA method NCFM (Wang et al., 2025) (top) employs unconstrained pixel-wise optimization, treating each time step of every sensor channel as an independent discrete variable. By ignoring intrinsic spatial coupling and continuous-time physical processes, this approach yields uncorrelated high-frequency spikes as degenerate solutions, resulting in suboptimal source ConvNet accuracy (green) and transfer failure on ResNet (red). In contrast, CHESS (bottom) constrains synthesis to a structured manifold by explicitly enforcing strong inter-channel correlations and temporal smoothness, ensuring stable optimization and robust cross-architecture generalization.

formal guaranties for smoothness and stability. Extensive experiments across diverse sensing modalities, downstream network architectures, and temporal resolutions, demonstrate that CHESS consistently outperforms state-of-the-art dataset condensation paradigms (i.e., coreset selection and dataset distillation) under extreme compression ratios (up to $133\times$) for each synthetic sample, while exhibiting robust cross-architecture generalization and zero-shot adaptation to unseen sampling resolutions.

## 2. Preliminaries

### 2.1. Continuous-Time Trajectory Model

we adopt a *trajectory-centric* modeling view, treating real-world multivariate signals as discretely-sampled observations of an underlying continuous trajectory. The primary modeling object is the continuous function itself, rather than its samples.

Formally, let $\mathcal{D}_{real} = \{(\mathbf{X}_i^{real}, y_i)\}_{i=1}^{|\mathcal{D}_{real}|}$ denote a large-scale dataset for classification task, where each $\mathbf{X}_i^{real} \in \mathbb{R}^{N \times S}$ consists of $N$ temporal samples collected through $S$ sensing channels. We assume each observation is associated with a latent continuous trajectory $\mathbf{h}_i(t) \in \mathbb{R}^S$, defined over a finite interval $t \in [0, \mathcal{T}]$, which captures the function-level evolution of the underlying physical or dynamical process, independent of any particular sampling grid.

To formalize this abstraction, we adopt the standard continuous-time dynamical system (CTDS) model (Drazin, 1992; Khalil & Grizzle, 2002). Concretely, we assume the existence of a latent state $\mathbf{x}(t) \in \mathbb{R}^d$, with its intrinsic dimensionality $d \ll NS$, evolving according to

$$\dot{\mathbf{x}}(t) = \mathbf{f}\left(\mathbf{x}(t), t\right), \qquad \mathbf{h}(t) = \mathbf{g}\left(\mathbf{x}(t), t\right) \in \mathbb{R}^S, \quad (1)$$

with $\mathbf{f} : \mathbb{R}^d \times \mathbb{R} \to \mathbb{R}^d$ governing the continuous-time dynamics, and $\mathbf{g} : \mathbb{R}^d \times \mathbb{R} \to \mathbb{R}^S$ mapping the latent state to channel-wise observations.

The observed discrete signal $\mathbf{X}_i^{real}$ is obtained by sampling $\mathbf{h}_i(t)$ at time points $\{t_n\}_{n=0}^{N-1} \subset [0, \mathcal{T}]$:

$$\mathbf{X}_i^{real}[n, :] = \mathbf{h}_i(t_n), \qquad n = 0, 1, \dots, N-1, \quad (2)$$

where the sampling grid $\{t_n\}_{n=0}^{N-1}$ and sampling frequency $f_s$ (i.e., $t_n = n/f_s$) may vary across sensing settings.

## 2.2. Objective for Synthetic Trajectory Learning

Although CHESS parameterizes synthesis at the continuous-trajectory level, optimization is necessarily conducted through an objective defined on discretely sampled realizations. We instantiate this objective via *distribution matching* (DM) (Zhang et al., 2024; Wang et al., 2025), which aligns real data $\mathcal{D}_{real}$ and synthetic data $\mathcal{D}_{syn}$ by matching their conditional feature statistics in a latent manifold induced by the encoder $\phi_\theta$. DM provides a stable, efficient, and architecture-agnostic alignment strategy without introducing bi-level optimization (Zhang et al., 2024).

Formally, the optimal synthetic dataset $\mathcal{D}_{syn}^*$ is obtained by minimizing the distance between the mean feature vectors of the real and synthetic sets for each class:

$$\underset{\mathcal{D}_{syn}}{\arg\min} \, \mathbb{E}_\theta \left\| \mu_\theta(\mathcal{D}_{real}) - \mu_\theta(\mathcal{D}_{syn}) \right\|^2, \quad (3)$$

$$\text{where} \quad \mu_\theta(\mathcal{D}) = \underset{\mathbf{x} \in \mathcal{D}}{\mathbb{E}}[\phi_\theta(\mathbf{x})].$$

Here, $\mathbf{X}^{syn} \in \mathcal{D}_{syn}$ denotes a synthetic example.

## 2.3. Stone-Weierstrass Theorem

The Stone-Weierstrass Theorem (Rudin, 1976) guarantees that any continuous function defined on a closed interval can be uniformly approximated by a polynomial. Formally, for any scalar-valued continuous trajectory $u(t)$, and error tolerance $\epsilon > 0$, there exists a polynomial $P(t)$ such that:

$$\sup_t |u(t) - P(t)| < \epsilon. \quad (4)$$

Motivated by this theorem, we adopt a polynomial parameterization, which is theoretically guaranteed to approximate trajectories with arbitrary precision.

## 2.4. Chebyshev Polynomials of the First Kind

The Chebyshev polynomials of the first kind (Mason & Handscomb, 2002), denoted as $\{T_k(t)\}_{k=0}^\infty$, form a sequence of orthogonal polynomials defined on the interval $t \in [-1, 1]$. They satisfy the following three-term recurrence relation:

$$T_{k+1}(t) = 2tT_k(t) - T_{k-1}(t), \quad (5)$$

with initial conditions $T_0(t) = 1$ and $T_1(t) = t$. Equivalently, these polynomials can be defined in terms of trigonometric functions as $T_k(t) = \cos(k \arccos t)$.

A key property of Chebyshev polynomials is their orthogonality with respect to the weight function $w(t) = (1 - t^2)^{-1/2}$. In approximation theory, this leads to their renowned *minimax* property (Mason & Handscomb, 2002): among all monic polynomials of degree $k$, the scaled Chebyshev polynomial $2^{1-k}T_k(t)$ achieves the smallest maximum absolute value on $[-1, 1]$. This optimality strictly controls the oscillation amplitude across the entire domain, effectively mitigating the unbounded fluctuations (Runge's phenomenon (Runge et al., 1901)) near interval boundaries.

# 3. Method

## 3.1. Low-Rank Spatial Modeling

Real-world continuous signals are typically generated by underlying physical or dynamical processes with a limited number of intrinsic degrees of freedom (DoF). Although the observed signal $\mathbf{X} \in \mathbb{R}^{N \times S}$ may appear high-dimensional due to multiple measurement channels, these channels are typically not independent owing to the correlated projections of the same latent processing. This inherent coupling across channels implies that the resulting signal matrix exhibits strong linear dependence and lies close to a low-dimensional subspace.

Motivated by this structural property, we model the synthetic signal generation process via an explicit low-rank factorization (Eckart & Young, 1936):

$$\mathbf{X}^{syn} = \mathbf{U}_{time} \mathbf{\Sigma} \mathbf{V}_{space}^\top, \quad (6)$$

where $\mathbf{U}_{time} \in \mathbb{R}^{N \times R}$ and $\mathbf{V}_{space} \in \mathbb{R}^{S \times R}$ characterize the temporal evolution and spatial channel dependencies, respectively, weighted by the singular values in $\mathbf{\Sigma}$. By strictly constraining the rank $R$, we enforce an regularization ensuring that the synthesized channels exhibit coherent activation patterns, effectively preventing the generation of uncorrelated noise or physically impossible channel combinations.

## 3.2. Piecewise Chebyshev Temporal Learning

Directly optimizing the discrete temporal basis $\mathbf{U}_{time}$ ignores the intrinsic continuity of the underlying continuous-

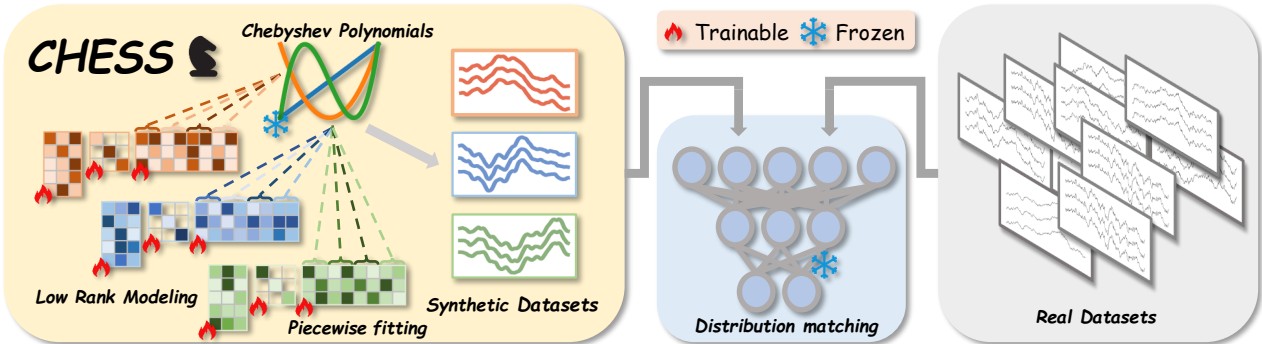

*Figure 2.* Schematic overview of the proposed CHESS framework. Departing from discrete pixel optimization, we parameterize synthetic data as continuous trajectories via *Low-Rank Modeling* and *Piecewise Chebyshev Fitting*. These structural priors enforce spatial coherence across channels and temporal smoothness, respectively, while parameters are learned by aligning feature distributions with real datasets.

time dynamics. Such point-wise optimization often over-fits the specific training architecture, leading to poor cross-model generalization and results in abrupt, unrealistic spikes rather than smooth signal trajectories (as shown in Fig. 1).

To address this, a conceptually natural way is to parameterize the temporal basis vectors in $\mathbf{U}_{time} \in \mathbb{R}^{N \times R}$ as $R$ continuous functions $\{u_r(t)\}_{r=1}^R$ rather than independent discrete points. However, modeling continuous functions is numerically challenging in discrete learning pipelines. A compromise is to approximate $\{u_r(t)\}_{r=1}^R$ using *polynomials*, as supported by *Stone-Weierstrass theorem* (Rudin, 1976) (Section 2.3). Nevertheless, approximating each long sequence with a single high-degree polynomial inevitably induces Runge's phenomenon (Runge et al., 1901), leading to boundary oscillations. To mitigate this, we employ a piecewise Chebyshev fitting strategy to ensure stable and smooth approximation. The total temporal dimension $N$ is partitioned into $M$ segments of length $L$. Let $\mathbf{B} \in \mathbb{R}^{(D+1) \times L}$ denote the fixed Chebyshev basis matrix. Its entries are constructed by evaluating Chebyshev polynomials on the normalized temporal domain[2] $[-1, 1]$:

$$\mathbf{B}_{i,l} = T_i\left(\frac{2l}{L-1} - 1\right), \qquad (7)$$

$$\text{where} \quad 0 \le i \le D, \ 0 \le l \le L-1,$$

where $T_k(x) = \cos(k \arccos x)$ is the Chebyshev polynomial of degree $k$. We reconstruct the temporal basis by learning the coefficient matrices $\{\mathbf{C}_m\}_{m=1}^M$, where $\mathbf{C}_m \in \mathbb{R}^{R \times (D+1)}$ parameterizes the trajectory within the $m$-th segment. The local basis for the segment is given by $\mathbf{U}_m = (\mathbf{C}_m \mathbf{B})^\top$, and the global basis is synthesized by matrix concatenation:

$$\mathbf{U}_{time} = \begin{bmatrix} \mathbf{U}_1 & \cdots & \mathbf{U}_M \end{bmatrix}^\top$$
$$= \begin{bmatrix} \mathbf{C}_1 \mathbf{B} & \cdots & \mathbf{C}_M \mathbf{B} \end{bmatrix}^\top. \qquad (8)$$

---

[2]Specifically, the local time domain of each segment is linearly rescaled to the canonical interval $[-1, 1]$ for Chebyshev evaluation.

This parameterization ensures that although the optimization objective is discrete, we are essentially optimizing $R$ continuous trajectories[3] $\{u_r(t)\}_{r=1}^R$, which acts as a structural denoiser against local spikes. Beyond its construction, this polynomial parameterization provides a rigorous theoretical guarantee on the smoothness of the synthesized signal trajectories, as formalized below:

**Proposition 1** (Derivative Control). *For any synthesized trajectory $u_r(t)$ within a segment $I_m$ of duration $\Delta T$ (corresponding to $L$ discrete samples), parameterized by polynomials of degree $D$, $\forall t \in I_m$, the temporal derivative is strictly bounded by:*

$$\|u_r'(t)\|_{L^\infty(I_m)} \le \frac{2}{\Delta T} D^2 \|u_r(t)\|_{L^\infty(I_m)}. \qquad (9)$$

*Proof.* Please refer to Appendix A.1. $\square$

While Proposition 1 constrains the individual basis functions, the final synthesized data $\mathbf{X}^{syn}$ is generated via the low-rank decomposition $\mathbf{X}^{syn} = \mathbf{U}_{time} \mathbf{\Sigma} \mathbf{V}_{space}^\top$ (Eq. 6). This matrix multiplication implies that synthesized trajectory in each channel is essentially a linear combination of the temporal basis functions $\{u_r(t)\}_{r=1}^R$. Therefore, it is crucial to verify that the smoothness property is preserved for the whole synthesized sample $\mathbf{X}^{syn}$:

**Lemma 1** (Bound Preservation for Linear Combinations). *Let $h(t) = \sum_r w_r u_r(t)$ be any linear combination of the trajectories on a segment $I_m$ of duration $\Delta T$, $h(t)$ strictly satisfies the derivative bound: $\|h'(t)\|_{L^\infty(I_m)} \le \frac{2}{\Delta T} D^2 \|h(t)\|_{L^\infty(I_m)}$.*

*Proof.* Please refer to Appendix A.2. $\square$

---

[3]Note that while each segment is individually smooth, we explicitly relax strict continuity constraints (e.g., $C^0$ value continuity or $C^1$ derivative continuity) at segment boundaries. As detailed in Appendix E.1, enforcing hard boundary constraints induces optimization instability and degrades performance.

This lemma serves as a strong structural inductive bias. Unlike unconstrained point-wise optimization, which permits arbitrary fluctuations, our framework imposes a hard constraint on the signal's rate of change proportional to $D^2$ within each segment. By leveraging the *minimax* property of Chebyshev polynomials to minimize the amplitude norm (see Appendix E.2 for a detailed discussion), this bound guarantees that the synthesized data remains physically meaningful. Our piecewise segmentation strategy plays a key role in making this derivative bound effective in practice. If a single global polynomial were employed for the entire sequence, the required degree $D$ would scale linearly with sequence length, causing the derivative bound $D^2$ to explode and lose its constraining power.

Chebyshev polynomials, and more broadly, structured function approximation, have been explored in prior techniques. For example, ChebNet (Defferrard et al., 2016) employs Chebyshev polynomials as spatial filters to approximate functions of the graph Laplacian on discrete graphs, while KAN (Liu et al., 2025) reparameterizes the *model* space to enhance function approximation capacity. Our approach is fundamentally different in *what* is being constrained. Rather than improving model expressiveness or designing more powerful operators, CHESS constrains the *data* parameter space itself. By representing trajectories through a Chebyshev temporal basis, we explicitly restrict synthesis to a physically meaningful function manifold.

### 3.3. Manifold-Aware Least Squares Initialization

We initialize parameters by projecting each real signal sample $\mathbf{X}^{real}$ onto the proposed manifold. We first decompose the real sample in each class using truncated SVD (Eckart & Young, 1936) to extract the components $\hat{\mathbf{U}}$, $\hat{\boldsymbol{\Sigma}}$, and $\hat{\mathbf{V}}$, directly assigning $\mathbf{V}_{space} \leftarrow \hat{\mathbf{V}}$ and $\boldsymbol{\sigma} \leftarrow \text{diag}(\hat{\boldsymbol{\Sigma}})$.

The challenge lies in determining the initial coefficient matrices $\{\mathbf{C}_m\}_{m=1}^M$ that best approximate the discrete temporal basis $\hat{\mathbf{U}}$. Let $\hat{\mathbf{U}}_m \in \mathbb{R}^{L \times R}$ denote the $m$-th row-segment of $\hat{\mathbf{U}}$ corresponding to the $m$-th segments. We formulate the initialization as a segment-wise linear least-squares (Penrose, 1955) problem. For each segment $m$, we solve:

$$\min_{\mathbf{C}_m} \left\| (\mathbf{C}_m \mathbf{B})^\top - \hat{\mathbf{U}}_m \right\|_F^2. \tag{10}$$

Leveraging the Moore-Penrose pseudo-inverse $\mathbf{B}^\dagger = \mathbf{B}^\top (\mathbf{B}\mathbf{B}^\top)^{-1}$, we derive the closed-form solution for each segment's coefficients: $\mathbf{C}_m = \hat{\mathbf{U}}_m^\top \mathbf{B}^\dagger$. This analytical initialization ensures that optimization starts from a physically meaningful manifold.

### 3.4. Learning and Training

Following recent DM paradigm in DANCE (Zhang et al., 2024) and NCFM (Wang et al., 2025), we adopt the same *pretrain-then-align* strategy, utilizing the intermediate states of a pre-trained expert model to guide the synthesis process. As shown in Fig. 2, unlike standard DD methods that update raw signal observations, we back-propagate the gradients through the differentiable reconstruction module to explicitly optimize the latent manifold parameters: the spatial basis $\mathbf{V}_{space}$, the set of temporal coefficient matrices $\{\mathbf{C}_m\}_{m=1}^M$, and the singular values $\boldsymbol{\sigma}$. The pseudocode of CHESS can be referred to Alg. 1 of Appendix C.

### 3.5. Zero-Shot Resolution Adaptation

A distinct advantage of CHESS is that it decouples the representation from any specific sampling grid. Conventional dataset condensation methods produce fixed tensors. As a result, adapting them to downstream models with different sampling rates requires manual interpolation or downsampling, which inevitably introduces information loss and signal distortion.

In contrast, CHESS enables resolution-agnostic adaptation without any redistillation. Since the learned coefficients $\{\mathbf{C}_m\}_{m=1}^M$ describe the continuous trajectory of the temporal basis $u_r(t)$, we can generate synthetic data at an arbitrary target sampling rate simply by re-evaluating the fixed Chebyshev basis matrix. Specifically, to synthesize data on a new length $L'$, we construct a new basis matrix $\mathbf{B}' \in \mathbb{R}^{(D+1) \times L'}$ by sampling the Chebyshev polynomials at the new timestamps. The adapted temporal basis $\mathbf{U}'_{time}$ is then analytically reconstructed via:

$$\mathbf{U}'_{time} = \begin{bmatrix} \mathbf{C}_1 \mathbf{B}' & \cdots & \mathbf{C}_M \mathbf{B}' \end{bmatrix}^\top. \tag{11}$$

The final synthetic data is obtained by $\mathbf{X}^{syn'} = \mathbf{U}'_{time} \boldsymbol{\Sigma} \mathbf{V}_{space}^\top$. This flexibility enables a *distill once, deploy anywhere* fashion, where a single synthesis can serve diverse downstream applications, regardless of their specific hardware configurations or sampling rates.

### 3.6. Storage and Compression Ratio Analysis

The baseline storage footprint for a raw synthetic sample is denoted as $NS$. In our framework, we absorb the trained singular values $\boldsymbol{\sigma}$ into the $\mathbf{V}_{space}$, representing each sample via a spatial basis $\mathbf{V}_{space} \in \mathbb{R}^{S \times R}$ and $M$ coefficient matrices $\{\mathbf{C}_m\}_{m=1}^M$. The compression ratio $\gamma$ for each sample is defined as:

$$\gamma = \frac{NS}{SR + RM(D+1)}. \tag{12}$$

Notably, the fixed Chebyshev basis matrix is excluded from the storage budget. As a deterministic mathematical construct, it remains constant across all samples and classes.

**Remark:** Unlike prior DD methods that strictly trade a fixed budget per class (BPC) to gain more samples per class

(SPC) (Kim et al., 2022b;a; Wei et al., 2023; Shin et al.), our work marks a *paradigm shift* driven by physical signal priors. We view the resulting high compression efficiency as an *ancillary benefit* of this shift. Therefore, we do not need to compromise or maneuver to exchange budget for sample count.

# 4. Experiments

To validate the generality of our continuous modeling paradigm, we evaluate CHESS on six sensor-based testbeds spanning diverse sensing modalities, each representing an instance of continuous physical signal generation observed through different measurement. The experiments cover classification performance (Sec. 4.1), cross-architecture generalization (Sec. 4.2), zero-shot resolution adaptation (Sec. 4.3), and ablation on basis functions and initialization (Sec. 4.4), followed by qualitative visualizations (Sec. 4.5).

**Datasets.** We conduct experiments on six sensor datasets spanning diverse sensing modalities, including four WiFi CSI datasets: three are self-built (i.e., ActR, MeR, FacT), as well as the public NTU-Fi dataset (Yang et al., 2023), the wearable sensor dataset PAMAP2 (Reiss & Stricker, 2012), and the widely used smartphone sensor-based UCI-HAR (Anguita et al., 2013) dataset. See Appendix D.2 for more details of datasets.

**Baselines.** To make a comprehensive comparison, we choose two dominant lines of prior work for dataset condensation: *i*) Coreset-based methods, including Random, Kmeans (Ahmed et al., 2020), Herding (Welling, 2009) and K-Center (Sener & Savarese, 2018); *ii*) Dataset distillation methods, including Gradient Matching (DC) (Zhao et al., 2021), Trajectory Matching (MTT) (Cazenavette et al., 2022), and Distribution Matching (DANCE) (Zhang et al., 2024), and NCFM (Wang et al., 2025). Notably, both DANCE and NCFM adopt the same DM objective with ours. This choice makes a controlled comparison that better highlights the effect of our paradigm itself, rather than differences in learning objective. See Appendix D.3 for the baseline implementations and Appendix D.5 for the detailed hyperparameters .

**Backbones.** We adopt ConvNet (LeCun et al., 2002) as the backbone $\phi_\theta$, and evaluate the generalization ability of distilled datasets on various backbones, including CNN backbones ResNet (He et al., 2016), AlexNet (Krizhevsky et al., 2012), ShuffleNet (Zhang et al., 2018), MobileNet (Howard et al., 2017) and a Transformer (Vaswani et al., 2017) backbone THAT (Li et al., 2021). To control complexity-induced impacts, all networks take the same three-layer architecture (above 90% accuracy). See Appendix D.4 for more details.

**Experimental Setup.** We repeat the distillation process twice for each method. The resulting synthetic datasets are evaluated five times using different random seeds. Results are reported as mean accuracy with standard deviation. See Appendix D.1 for details.

## 4.1. Main Results

As shown in Table 1, CHESS achieves state-of-the-art performance in both accuracy and compression across all six datasets. This success stems from our paradigm shift: by learning from a structured low-rank polynomial manifold, rather than the raw discrete signal space, CHESS captures the essential dynamics while discarding noise. For data efficiency, CHESS achieving a stunning $133.3\times$ compression on high sampling rate signals (e.g., ActR, MeR, and FacT). The main reason is that CHESS is able to exploit the significant temporal redundancy of data having high sampling rate. Even on lower-rate sensor datasets like PAMAP2 and UCI-HAR, CHESS still achieves a stable compression ratio between $6.7\times$ and $6.9\times$. For accuracy, our structural constraint acts as a powerful denoiser, preventing performance collapse even under these aggressive compression ratios. By guiding the model toward solutions that align with the intrinsic continuous generation process of the sensor data, CHESS maintains superior accuracy. On average, CHESS improves accuracy by approximately 6.4% over the strongest baseline. On complex ActR dataset at the SPC=5, where CHESS attains 71.0%, outperforming the runner-up DC method by a substantial margin of 11.2%.

## 4.2. Cross-Architecture Generalization

To verify if synthesized data captures intrinsic semantics rather than overfitting, we evaluate transferability across diverse architectures ranging from standard CNNs to the Transformer-based THAT on MeR and UCI-HAR as shown in Table 2. Conventional methods struggle with architecture shifts. For instance, DANCE and NCFM drop below 50% on ShuffleNet within the MeR dataset, and baselines like MTT fail to generalize to THAT likely due to architecture-specific artifacts. In contrast, CHESS remains robust on the complex MeR dataset as it outperforms DANCE on ShuffleNet by 16.6% and achieves the highest accuracy of 67.4% on the Transformer backbone. While CHESS slightly trails DC on the simpler UCI-HAR dataset where DC achieves 64.0% compared to 62.9% for CHESS on THAT, our method maintains significantly better stability across lightweight networks compared to other baselines. Ablation studies confirm our design choices. Removing Chebyshev Polynomials denoted as *w/o C.P* drops MobileNet accuracy by over 9% which proves its role in suppressing spikes. Similarly, removing the low-rank constraint denoted as *w/o SVD* consistently degrades performance, suggesting the importance of spatial priors. Please refer to Appendix D.7 for more cross-architecture generalizations experiments.

*Table 1.* Main results: Performance comparison of Coreset selection methods and DD methods across six sensor signal datasets. **SPC** denotes Samples Per Class. **Ratio (%)** indicates the condensation ratio, calculated as the size of the synthetic subset relative to the full training set. The column $\gamma$ represents the storage compression ratio of each sample synthesised by CHESS. Results are reported as classification accuracy (mean $\pm$ std) over 10 independent runs. Highlights indicate the best , second-best , and third-best results.

| Dataset | SPC | Ratio (%) | Coreset Selection | | | | Dataset Distillation | | | | | $\gamma$ | Full Data |
|---|---|---|---|---|---|---|---|---|---|---|---|---|---|
| | | | *Random* | *K-Means* | *Herding* | *K-Center* | *DC* | *MTT* | *DANCE* | *NCFM* | *CHESS* | | |
| **ActR** | 1 | 0.17 | $28.4_{\pm1.7}$ | $32.7_{\pm2.3}$ | $21.3_{\pm1.9}$ | $25.8_{\pm3.1}$ | $38.9_{\pm2.5}$ | $39.8_{\pm3.8}$ | $40.1_{\pm1.6}$ | $42.8_{\pm2.9}$ | $\mathbf{43.7}_{\pm2.9}$ | $133.3\times$ | $92.2_{\pm1.8}$ |
| | 5 | 0.88 | $39.3_{\pm2.2}$ | $53.8_{\pm1.8}$ | $38.1_{\pm3.5}$ | $33.4_{\pm0.9}$ | $59.8_{\pm1.7}$ | $57.1_{\pm3.1}$ | $56.9_{\pm2.6}$ | $56.9_{\pm1.9}$ | $\mathbf{71.0}_{\pm1.4}$ | | |
| | 10 | 1.77 | $46.2_{\pm0.8}$ | $53.4_{\pm1.5}$ | $39.7_{\pm2.4}$ | $38.3_{\pm3.2}$ | $63.2_{\pm1.9}$ | $64.8_{\pm2.7}$ | $64.4_{\pm1.4}$ | $63.9_{\pm2.1}$ | $\mathbf{74.7}_{\pm3.4}$ | | |
| | **Average** | | 38.0 | 46.6 | 33.0 | 32.5 | 54.0 | 53.9 | 53.8 | 54.5 | **63.1** | | |
| **MeR** | 1 | 0.17 | $32.7_{\pm1.2}$ | $40.9_{\pm2.5}$ | $43.6_{\pm1.8}$ | $29.7_{\pm3.3}$ | $43.2_{\pm1.6}$ | $51.6_{\pm2.9}$ | $47.2_{\pm1.4}$ | $49.4_{\pm2.7}$ | $\mathbf{52.3}_{\pm1.9}$ | $133.3\times$ | $92.3_{\pm3.1}$ |
| | 5 | 0.88 | $55.8_{\pm0.9}$ | $57.2_{\pm1.7}$ | $55.4_{\pm2.6}$ | $39.5_{\pm1.5}$ | $62.4_{\pm3.4}$ | $59.3_{\pm2.1}$ | $63.8_{\pm1.9}$ | $66.7_{\pm2.8}$ | $\mathbf{77.1}_{\pm1.5}$ | | |
| | 10 | 1.77 | $60.0_{\pm1.1}$ | $62.2_{\pm2.3}$ | $55.6_{\pm1.4}$ | $43.8_{\pm2.9}$ | $64.8_{\pm1.8}$ | $70.2_{\pm3.3}$ | $69.7_{\pm2.5}$ | $72.1_{\pm1.7}$ | $\mathbf{79.1}_{\pm1.8}$ | | |
| | **Average** | | 49.5 | 53.4 | 51.5 | 37.7 | 56.8 | 60.4 | 60.2 | 62.7 | **69.5** | | |
| **FacT** | 1 | 0.17 | $30.3_{\pm0.8}$ | $30.2_{\pm1.6}$ | $27.5_{\pm2.4}$ | $16.5_{\pm1.3}$ | $34.7_{\pm3.1}$ | $39.3_{\pm2.2}$ | $33.1_{\pm1.7}$ | $34.6_{\pm2.9}$ | $\mathbf{45.4}_{\pm2.2}$ | $133.3\times$ | $87.1_{\pm2.9}$ |
| | 5 | 0.88 | $42.4_{\pm1.5}$ | $41.2_{\pm2.1}$ | $29.3_{\pm1.2}$ | $27.7_{\pm2.7}$ | $48.2_{\pm1.9}$ | $46.2_{\pm3.2}$ | $45.1_{\pm2.3}$ | $51.3_{\pm1.8}$ | $\mathbf{62.1}_{\pm3.7}$ | | |
| | 10 | 1.77 | $49.2_{\pm0.7}$ | $47.1_{\pm1.9}$ | $35.3_{\pm2.8}$ | $37.8_{\pm1.1}$ | $52.1_{\pm2.5}$ | $55.5_{\pm1.6}$ | $58.0_{\pm3.0}$ | $56.3_{\pm2.0}$ | $\mathbf{65.4}_{\pm1.6}$ | | |
| | **Average** | | 40.6 | 39.5 | 30.7 | 27.3 | 45.0 | 47.0 | 45.4 | 47.4 | **57.6** | | |
| **NTU-Fi** | 1 | 0.64 | $32.2_{\pm1.2}$ | $41.3_{\pm2.8}$ | $47.4_{\pm1.9}$ | $39.0_{\pm2.5}$ | $62.4_{\pm1.7}$ | $68.2_{\pm3.3}$ | $68.9_{\pm2.1}$ | $68.4_{\pm1.5}$ | $\mathbf{71.7}_{\pm1.8}$ | $5.3\times$ | $94.1_{\pm3.2}$ |
| | 5 | 3.11 | $59.1_{\pm0.6}$ | $75.8_{\pm1.8}$ | $71.6_{\pm2.7}$ | $48.9_{\pm1.4}$ | $62.1_{\pm2.9}$ | $82.5_{\pm1.6}$ | $89.9_{\pm3.1}$ | $92.4_{\pm2.4}$ | $\mathbf{94.4}_{\pm4.5}$ | | |
| | **Average** | | 45.7 | 58.6 | 59.5 | 44.0 | 62.3 | 75.4 | 79.4 | 80.4 | **83.1** | | |
| **PAMAP2** | 1 | 0.05 | $22.4_{\pm0.1}$ | $23.1_{\pm0.7}$ | $33.3_{\pm0.5}$ | $24.2_{\pm0.4}$ | $44.2_{\pm0.4}$ | $38.9_{\pm1.5}$ | $45.2_{\pm0.9}$ | $42.1_{\pm0.4}$ | $\mathbf{55.6}_{\pm2.1}$ | $6.9\times$ | $97.2_{\pm0.4}$ |
| | 5 | 0.26 | $44.5_{\pm1.2}$ | $41.3_{\pm0.4}$ | $37.1_{\pm0.9}$ | $37.6_{\pm0.5}$ | $54.5_{\pm4.2}$ | $50.4_{\pm0.5}$ | $60.5_{\pm1.1}$ | $60.2_{\pm0.1}$ | $\mathbf{63.5}_{\pm1.7}$ | | |
| | 10 | 0.51 | $59.3_{\pm0.6}$ | $52.7_{\pm0.3}$ | $34.1_{\pm0.3}$ | $41.6_{\pm0.7}$ | $56.6_{\pm0.4}$ | $63.9_{\pm0.1}$ | $63.4_{\pm1.4}$ | $66.8_{\pm0.7}$ | $\mathbf{71.8}_{\pm0.3}$ | | |
| | **Average** | | 42.1 | 39.0 | 34.8 | 34.5 | 51.8 | 51.1 | 56.4 | 56.4 | **63.6** | | |
| **UCI-HAR** | 1 | 0.09 | $63.8_{\pm1.0}$ | $66.7_{\pm1.2}$ | $55.2_{\pm2.2}$ | $58.2_{\pm3.5}$ | $65.6_{\pm2.1}$ | $64.8_{\pm6.1}$ | $59.4_{\pm1.2}$ | $59.3_{\pm2.2}$ | $\mathbf{70.3}_{\pm2.3}$ | $6.7\times$ | $98.1_{\pm0.2}$ |
| | 5 | 0.46 | $67.7_{\pm1.1}$ | $71.2_{\pm1.4}$ | $61.7_{\pm0.8}$ | $61.9_{\pm1.8}$ | $79.5_{\pm6.1}$ | $73.4_{\pm1.6}$ | $77.2_{\pm3.1}$ | $78.9_{\pm2.4}$ | $\mathbf{80.8}_{\pm1.4}$ | | |
| | **Average** | | 65.8 | 69.0 | 58.5 | 60.1 | 72.6 | 69.1 | 68.3 | 69.1 | **75.6** | | |

DC

MTT

NCFM

CHESS (Ours)

*Figure 3.* Visual comparison of distilled samples of first 5 channels on MeR dataset. Baseline methods (DC, MTT, NCFM) are dominated by high-frequency adversarial spikes, indicating severe overfitting to the surrogate model. In contrast, CHESS produces smooth and realistic signals with clear patterns. It effectively removes noise and spikes by enforcing structural smoothness.

### 4.3. Zero-Shot Resolution Generalization

To assess the capability for zero-shot resolution adaptation, we conducted experiments on the igh sampling rate MeR dataset. The original signals possess a high temporal resolution of 2000 time steps. During the distillation phase, we downsampled these signals to a sequence length of 1000 to serve as the training source. We then evaluated the models on two distinct targets without redistillation, specifically a low-resolution test set decimated to a length of 200 and the original high-resolution test set of length 2000. Regarding the baselines, we employed time-domain linear interpolation and linear downsampling. In contrast, CHESS uses the analytical resampling mechanism detailed in Section 3.5. As shown in Table 3, discrete baselines suffer from interpolation artifacts, evidenced by DANCE degrading to 62.3% on 2000-step upsampling. By leveraging analytic polynomial resampling, CHESS avoids such losses and achieves 75.8% accuracy, representing a 12.5% improvement. This

*Table 2.* Cross-architecture performance comparison on MeR (SPC=5) and UCI-HAR datasets (SPC=1). W/o SVD: removes low-rank constraint; w/o C.P: removes Chebyshev fitting. Highlights indicate the **best** , second-best , and third-best results.

| Method | ResNet | AlexNet | ShuffleNet | MobileNet | THAT |
|---|---|---|---|---|---|
| **MeR** | | | | | |
| DC | $70.8_{\pm1.6}$ | $74.3_{\pm0.8}$ | $62.0_{\pm2.7}$ | $69.4_{\pm0.0}$ | $66.7_{\pm1.2}$ |
| MTT | $47.5_{\pm1.9}$ | $59.8_{\pm0.9}$ | $48.2_{\pm9.7}$ | $52.0_{\pm4.7}$ | $66.9_{\pm1.3}$ |
| DANCE | $58.3_{\pm3.6}$ | $64.1_{\pm2.3}$ | $48.6_{\pm14.8}$ | $50.9_{\pm9.2}$ | $66.4_{\pm0.6}$ |
| NCFM | $52.9_{\pm3.1}$ | $62.9_{\pm4.7}$ | $49.8_{\pm11.7}$ | $50.5_{\pm15.1}$ | $66.6_{\pm1.1}$ |
| CHESS w/o SVD | $72.5_{\pm1.2}$ | $75.2_{\pm1.1}$ | $64.5_{\pm1.1}$ | $71.1_{\pm1.7}$ | $67.1_{\pm0.3}$ |
| CHESS w/o C.P | $70.3_{\pm1.6}$ | $69.1_{\pm1.7}$ | $61.1_{\pm2.1}$ | $62.2_{\pm1.6}$ | $64.2_{\pm3.2}$ |
| **CHESS** | $73.2_{\pm1.7}$ | $75.8_{\pm1.3}$ | $65.2_{\pm3.8}$ | $71.6_{\pm2.5}$ | $67.4_{\pm1.6}$ |
| **UCI-HAR** | | | | | |
| DC | $63.4_{\pm1.9}$ | $63.5_{\pm1.5}$ | $60.5_{\pm2.7}$ | $62.6_{\pm0.8}$ | $64.0_{\pm4.8}$ |
| MTT | $64.8_{\pm1.7}$ | $66.4_{\pm0.9}$ | $56.9_{\pm1.3}$ | $61.4_{\pm1.1}$ | $63.2_{\pm7.1}$ |
| DANCE | $59.2_{\pm3.7}$ | $60.7_{\pm1.2}$ | $45.1_{\pm8.7}$ | $44.6_{\pm2.1}$ | $61.7_{\pm3.0}$ |
| NCFM | $55.6_{\pm2.9}$ | $61.8_{\pm2.9}$ | $46.0_{\pm8.9}$ | $44.4_{\pm3.0}$ | $63.2_{\pm3.4}$ |
| CHESS (w/o SVD) | $64.5_{\pm1.3}$ | $67.8_{\pm2.3}$ | $58.5_{\pm2.1}$ | $65.5_{\pm2.3}$ | $63.3_{\pm1.4}$ |
| CHESS (w/o C.P) | $60.3_{\pm2.9}$ | $60.1_{\pm5.7}$ | $52.2_{\pm1.6}$ | $62.2_{\pm1.2}$ | $60.2_{\pm6.2}$ |
| **CHESS** | $66.4_{\pm1.7}$ | $68.2_{\pm2.1}$ | $61.6_{\pm1.9}$ | $66.4_{\pm1.3}$ | $62.9_{\pm3.3}$ |

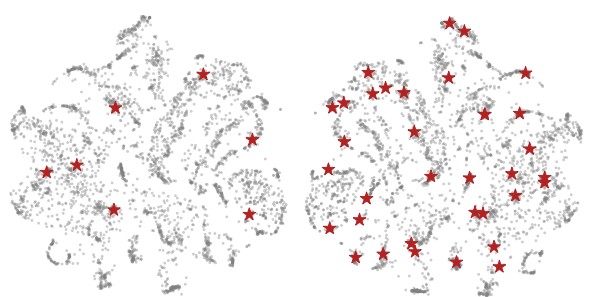

*Figure 4.* Feature distributions of the MeR dataset. Grey dots represent the original data, while red stars indicate distilled samples generated by CHESS. Left: SPC=1; Right: SPC=5. The distilled samples effectively cover the underlying distribution of the original dataset, ensuring both physical meaningful and semantic diversity.

*Table 3.* Zero-shot resolution adaptation performance on MeR dataset (SPC=5). Models are distilled at a sequence length of $L = 1000$ and evaluated on $L = 200$ and $L = 2000$ without redistillation. Highlights indicate the **best** results.

| Length | 200 | **1000 (Train)** | 2000 |
|---|---|---|---|
| DANCE | $66.3 \pm 0.7$ | $65.2 \pm 0.1$ | $62.3 \pm 0.9$ |
| NCFM | $64.2 \pm 0.8$ | $63.4 \pm 1.0$ | $62.1 \pm 2.0$ |
| **CHESS** | $76.9 \pm 0.2$ | $76.2 \pm 0.2$ | $75.8 \pm 1.1$ |

robustness extends to downsampling to length 200, where CHESS maintains 76.9% accuracy and consistently outperforms baselines by preserving signal semantics across scales. For more zero-shot resolution generalization experiments, please refer to Appendix D.8.

## 4.4. Ablation on Basis Functions and Initialization.

In this section, we analyze the impact of different polynomial bases and initialization strategies on MeR datasaet in Table 4 where the synthetic data is distilled using a source ConvNet and evaluated on a target ResNet. The results reveal two critical insights. First, initialization is decisive as random initialization fails to generate meaningful signals. Our method requires a physically grounded warm start via our Least Squares projection initialization as shown in Section 3.3. Second, weighted orthogonality of polynomials (Szeg, 1939) plays a pivotal role in optimization stability. The standard monomial basis *Poly* underperforms significantly due to severe collinearity, which leads to an ill-conditioned landscape. In contrast, polynomial basis function defined by weighted orthogonality (e.g., *Chebyshev*, *Legendre*, and *Hermite*) naturally decorrelate the learned coefficients. This structural advantage ensures superior robustness, yielding over 73% accuracy on ResNet.

**Sensitivity of polynomial degree.** Appendix D.9 presents a sensitivity analysis of the polynomial degree $D$. The optimal accuracy is consistently attained at low degrees ($D \in [2, 3]$) across all datasets, beyond which performance either saturates or degrades due to the Runge phenomenon. These results confirm that the observed gains arise from the orthogonal polynomial parameterization rather than from an increase in model capacity.

*Table 4.* Ablation study on polynomial basis functions and initialization strategies. We report the classification accuracy on ConvNet and ResNet. Highlights indicate the **best** results.

| Method | Least Square | | Random | |
|---|---|---|---|---|
| | ConvNet | ResNet | ConvNet | ResNet |
| Poly | $65.3 \pm 0.4$ | $57.1 \pm 0.2$ | $14.2 \pm 0.7$ | $14.3 \pm 0.3$ |
| Hermite | $77.1 \pm 0.5$ | $73.1 \pm 0.4$ | $15.1 \pm 1.5$ | $15.0 \pm 0.2$ |
| Legendre | $76.9 \pm 1.4$ | $73.2 \pm 0.9$ | $14.7 \pm 0.8$ | $14.3 \pm 0.2$ |
| Chebyshev | $77.1 \pm 1.5$ | $73.2 \pm 1.7$ | $14.5 \pm 0.3$ | $14.4 \pm 0.2$ |

## 4.5. Visualization

**Visualization of Synthetic Samples.** We plot the samples for the first five channels of the distilled MeR dataset, as shown in Fig. 3. Methods relying on unconstrained optimization such as DC, MTT, NCFM produce waveforms characterized by chaotic, high-frequency spikes. Conversely, CHESS produces clean, continuous trajectories with distinct structures. For more visualization of samples synthesized by CHESS, please refer to the Appendix F.2.

**Visualization of T-SNE.** (Maaten & Hinton, 2008) Fig. 4 visualizes the last layer feature space embedding of the MeR dataset. We observe that the distilled samples (red stars) are faithfully embedded within the high-density regions of the

original data manifold (grey dots). This alignment confirms that CHESS successfully captures the intrinsic semantic structure of the sensor signals. For more visualization of t-SNE, please refer to the Appendix F.1.

## 5. Limitations

CHESS is explicitly tailored for multivariate CTDS signals governed by shared continuous-time physical processes, leveraging the strong spatial correlations inherent in simultaneous observations. Consequently, CHESS may not directly generalize to unstructured multivariate time-series, such as financial or server data, where variables often evolve stochastically with weak inter-dependencies. Furthermore, while our current evaluation focuses exclusively on classification tasks, extending this *function-first* paradigm to other time-series tasks, such as forecasting and regression, remains a promising direction for future research.

## 6. Conclusion

In this work, we introduce CHESS to resolve the misalignment between discrete condensation and continuous CTDS trajectories. By shifting optimization to a manifold defined by low-rank spatial coherence and piecewise Chebyshev temporal continuity, CHESS eliminates high-frequency artifacts and ensures physically meaningful trajectories. Evaluations across six testbeds establish CHESS as the state-of-the-art, achieving extreme compression with robust generalization. Unlike fixed-tensor baselines, its continuous parameterization enables unprecedented zero-shot adaptation to arbitrary sampling resolutions, offering a scalable foundation for resource-constrained sensing intelligence.

## Impact Statement

This paper presents work whose goal is to advance the field of Machine Learning. There are many potential societal consequences of our work, none which we feel must be specifically highlighted here.

## Acknowledgements

This work was supported by the National Natural Science Foundation of China (Grant No. 62476053), NSFC Key International (Regional) Joint Research Program (Grant No. W2511067), the Green Buildings Innovation Cluster 2.0 Challenge Call for Decarbonisation (GBIC R&D/2025/4), and the Fundamental Research Funds for the Central Universities (Grant No. ZYGX2024J003).

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

# A. Proof of Theoretical Analysis

## A.1. Proof of Proposition 1

Before proving Proposition 1, we first introduce the classic Markov Brothers' Inequality (Markov, 1892), which provides a bound on the derivatives of polynomials on a standardized interval.

**Theorem 1** (Markov Brothers' Inequality). *Let $\mathbb{P}_D$ denote the vector space of polynomials with real coefficients and degree at most $D$. For any polynomial $P \in \mathbb{P}_D$ defined on the canonical interval $\tau \in [-1, 1]$, the supremum norm of its first derivative is bounded by:*

$$\|P'(\tau)\|_{L^\infty([-1,1])} \leq D^2 \|P(\tau)\|_{L^\infty([-1,1])}. \tag{13}$$

Building upon Theorem 1, we derive the derivative bound for our piecewise parameterization in the physical time domain.

**Proposition 1** (Derivative Control). *For any synthesized trajectory $u_r(t)$ within a segment $I_m$ of length $\Delta T$, parameterized by polynomials of degree $D$, the temporal derivative is strictly bounded by:*

$$\|u_r'(t)\|_{L^\infty(I_m)} \leq \frac{2}{\Delta T} D^2 \|u_r(t)\|_{L^\infty(I_m)}, \quad \forall t \in I_m. \tag{14}$$

*Proof.* Let $I_m = [t_0, t_0 + \Delta T]$ denote the physical time interval for the $m$-th segment. We define an affine mapping between the physical time $t \in I_m$ and the canonical reference time $\tau \in [-1, 1]$:

$$\tau(t) = \frac{2}{\Delta T}(t - t_0) - 1, \quad \text{with} \quad \frac{d\tau}{dt} = \frac{2}{\Delta T}. \tag{15}$$

Let $P(\tau)$ represent the polynomial trajectory in the reference domain such that $P(\tau(t)) = u_r(t)$. By the Chain Rule, the relationship between the derivatives in the two domains is:

$$\frac{du_r}{dt} = \frac{dP}{d\tau} \cdot \frac{d\tau}{dt} = P'(\tau) \cdot \frac{2}{\Delta T}. \tag{16}$$

Applying the *Markov Brothers' Inequality* (Theorem 1) to $P(\tau)$ on $[-1, 1]$, we have:

$$\|P'(\tau)\|_{L^\infty([-1,1])} \leq D^2 \|P(\tau)\|_{L^\infty([-1,1])}. \tag{17}$$

Substituting the physical time quantities back into this inequality:

$$\|u_r'(t)\|_{L^\infty(I_m)} = \left\| P'(\tau) \cdot \frac{2}{\Delta T} \right\|_{L^\infty([-1,1])} \tag{18}$$

$$= \frac{2}{\Delta T} \|P'(\tau)\|_{L^\infty([-1,1])} \tag{19}$$

$$\leq \frac{2}{\Delta T} D^2 \|P(\tau)\|_{L^\infty([-1,1])} \tag{20}$$

$$= \frac{2}{\Delta T} D^2 \|u_r(t)\|_{L^\infty(I_m)}. \tag{21}$$

This concludes the proof. $\square$

## A.2. Proof of Lemma 1

**Lemma 1** (Bound Preservation for Linear Combinations). *Let $h(t) = \sum_r w_r u_r(t)$ be any linear combination of the trajectories on a segment $I_m$ of length $\Delta T$, where each $u_r(t)$ is a polynomial of degree at most $D$. Then $h(t)$ strictly satisfies the derivative bound:*

$$\|h'(t)\|_{L^\infty(I_m)} \leq \frac{2}{\Delta T} D^2 \|h(t)\|_{L^\infty(I_m)}. \tag{22}$$

*Proof.* The proof relies on the closure property of the polynomial vector space. Let $\mathbb{P}_D$ be the space of polynomials with degree at most $D$. By definition, the basis trajectories satisfy $u_r(t) \in \mathbb{P}_D$ for all $r$. Since vector spaces are closed under linear combinations, the synthesized signal $h(t)$ satisfies:

$$h(t) = \sum_r w_r u_r(t) \in \mathbb{P}_D. \tag{23}$$

This implies that $h(t)$ is itself a polynomial of some degree $d$, where $d \leq D$. Applying Proposition 1 directly to the polynomial $h(t)$, its derivative is bounded by its own degree $d$:

$$\|h'(t)\|_{L^\infty(I_m)} \leq \frac{2}{\Delta T} d^2 \|h(t)\|_{L^\infty(I_m)}. \tag{24}$$

Since the degree of the linear combination cannot exceed the maximum degree of the basis (i.e., $d \leq D$) and the square function is monotonic for non-negative numbers, it strictly follows that $d^2 \leq D^2$. Therefore:

$$\|h'(t)\|_{L^\infty(I_m)} \leq \frac{2}{\Delta T} D^2 \|h(t)\|_{L^\infty(I_m)}. \tag{25}$$

This confirms that the explicit smoothness regularization imposed on the temporal basis $\{u_r(t)\}$ extends to the entire spatial manifold of the synthesized data $\mathbf{X}^{syn}$. □

## B. Related work

### B.1. Continuous-Time Dynamical System Model

Real-world sensor data, ranging from wearable IMUs to wireless CSI, fundamentally represent discrete snapshots of underlying continuous-time dynamical systems (CTDS). In recent years, deep learning has revolutionized the analysis of such signals, establishing itself as the dominant paradigm for applications including human activity recognition (Wang et al., 2015; Li et al., 2021) and physiological monitoring (Maweu et al., 2021; Yang et al., 2019). Early methodologies primarily leveraged Convolutional Neural Networks (CNNs) (Wang et al., 2019) and Recurrent Neural Networks (RNNs) (Ding & Wang, 2019) to extract local spatio-temporal features and model sequential dependencies from sensor streams. Building on this foundation, the field has rapidly advanced towards more sophisticated architectures. Attention (Vaswani et al., 2017)-based mechanisms and Transformers (Li et al., 2021) have been widely adopted to capture long-range temporal correlations, while emerging continuous-time paradigms, such as Neural ODEs (Rubanova et al., 2019) and Controlled Differential Equations (Kidger et al., 2020) (CDEs), have enabled explicit modeling of the underlying physical dynamics and irregular sampling intervals. These progressions demonstrate a consistent trend towards developing increasingly high-fidelity representations of physical sensor trajectories.

### B.2. Dataset Distillation

Dataset Distillation (DD) (Wang et al., 2018) aims to condense large-scale datasets into compact synthetic counterparts while preserving their training efficacy. The prevailing paradigm primarily focuses on optimizing discrete observations (e.g., pixel values) via objectives such as gradient matching (Zhao et al., 2021), trajectory matching (Cazenavette et al., 2022), or distribution matching (Zhao & Bilen, 2023). However, these approaches inherently treat synthetic data as a collection of independent discrete variables, ignoring the underlying dynamical laws that govern real-world data. To the best of our knowledge, this paper is the first to incorporate explicit physical priors into the DD framework, leveraging these properties to enforce strict temporal smoothness and physical validity in continuous time dynamic signals.

### B.3. Parameterization of Dataset Distillation

The parameterization of the distilled dataset critically influences both performance and efficiency. To address the limitations of pixel-wise optimization, structural approaches such as decoder-based (Liu et al., 2022; Wei et al., 2023) and dictionary-based (Deng & Russakovsky, 2022) models were proposed to reduce dimensionality via latent codes or basis combinations. More recently, research has shifted towards compact representations, including Implicit Neural Representations (INRs) (Shin et al.) and Gaussian Splatting (Jiang et al., 2025), aiming to maximize sample counts under fixed storage budgets. Crucially, our work marks a fundamental paradigm shift from these prior methods (Kim et al., 2022b;a; Wei et al., 2023; Shin et al.), which typically treat distillation as a zero-sum game: strictly trading a fixed storage budget (Budget Per Class) to gain more samples (Samples Per Class). Instead of balancing this trade-off through budget manipulation, we approach synthesis through the lens of *physical signal priors*. By modeling data as continuous, low-rank trajectories, high compression efficiency emerges not as a calculated compromise, but as a natural *ancillary benefit* of the underlying mathematical structure. Consequently, our framework obviates the need to sacrifice signal fidelity for sample quantity, achieving scalability through rigorous trajectory modeling.

### B.4. Chebyshev Polynomials in Machine Learning

Chebyshev polynomials are a sequence of orthogonal polynomials known for their optimal approximation properties, particularly in minimizing the Runge phenomenon (Mason & Handscomb, 2002). In machine learning, they have been extensively applied in Graph Neural Networks (GNNs). ChebNet (Defferrard et al., 2016) utilizes Chebyshev expansion to approximate spectral graph filters, avoiding computationally expensive eigen-decompositions. In robotics, Agrawal et al. (Agrawal & Dellaert, 2021) introduce a novel continuous-time estimation framework based on Chebyshev pseudo-spectral parameterization that enables the joint recovery of robot state trajectories and control dynamics. In the context of signal processing, Chebyshev interpolation is the gold standard for representing smooth functions (Trefethen, 2019; Zhang et al., 2025; Wu, 2019).

## C. Implementation Details of CHESS

## D. Experimental Details

This section is organized to provide a comprehensive guide to reproducibility and extended empirical analysis. We first detail the experimental infrastructure, including the hardware specifications in Sec. D.1 and the statistics of the six sensor datasets in Sec. D.2. Next, we elaborate on the implementation details of baseline methods in Sec. D.3, backbone architectures in Sec. D.4, and specific hyperparameter configurations in Sec. D.5. Finally, we present supplementary experiments that further validate the robustness of CHESS, including comparison under standardized BPC settings in Sec. D.6, extended cross-architecture generalization in Sec. D.7, zero-shot resolution adaptation in Sec. D.8, hyperparameter analysis in Sec. D.9, and computational efficiency analysis in Sec. D.10.

### D.1. Experimental Environment

All experiments were conducted on a high-performance Linux server equipped with dual Intel Xeon Gold 6530 CPUs (totaling 64 physical cores and 128 threads, with a max frequency of 4.0 GHz) and two NVIDIA GeForce RTX 5090 GPUs (32GB VRAM each).

### D.2. Detail of Datasets

In this section, we detail the datasets employed for evaluation and provide their public sources to facilitate reproducibility. To provide a comprehensive evaluation of our proposed method, we utilize a total of six distinct CTDS signal datasets. Specifically, we constructed three proprietary high-sampling-rate WiFi Channel State Information (CSI) datasets ActR, MeR, and FacT which captured across diverse environmental settings. Complementing these, we incorporate three widely recognized public benchmarks: NTU-Fi, PAMAP2, and UCI-HAR, to verify the generalizability and robustness of our approach across different sensor modalities and standard community baselines.

D.2.1. PROPRIETARY WIFI CSI DATASETS

We collected data using a uniform hardware configuration deploying both Atheros 9590 and Intel 5300 Network Interface Cards (NICs) operating at 2.4GHz with a 20MHz bandwidth. The router was equipped with three antennas, each monitoring 10 subcarriers. To capture comprehensive motion patterns, we employed a sliding window approach with a duration of 4 seconds. With a sampling frequency of 1000 Hz, each resulting data sample possesses a shape of $2000 \times 30$. The specific environments are described as follows:

**ActR**: A complex indoor WiFi CSI dataset collected in a small-scale activity room cluttered with sundries. The dataset involves 16 participants (varying in gender and body type) performing 7 distinct activities (sit, pick, wave, jump, walk, run, and empty). The transmitter-receiver distance was set to 3 meters to capture high-quality CSI data amidst significant signal scattering.

**MeR**: A multipath-rich dataset constructed in a standard meeting room furnished with multiple tables and chairs. Similar to ActR, we recorded data from 16 subjects performing the same 7 types of actions, with the transceiver antennas placed at a height of 1 meter. This dataset, comprising over 4900 samples collected from our Atheros and Intel streams, provides a rigorous testbed for modeling human activities in scenarios dominated by static multipath reflections from office furniture.

**Algorithm 1** PyTorch-style Pseudocode of CHESS

```python
import torch
import torch.nn as nn
import numpy as np
from numpy.polynomial.chebyshev import chebvander
from numpy.linalg import pinv

class CHESS_Synthesizer(nn.Module):
    def __init__(self, rank, seg_len, degree, num_syn, channels):
        super().__init__()

        # 1. Precompute Chebyshev Basis (B) and Pseudo-Inverse (B_pinv)
        # Generate Chebyshev nodes in [-1, 1]
        x_grid = np.linspace(-1, 1, seg_len)
        # B: [seg_len, degree+1] -> Transpose to [degree+1, seg_len]
        basis = chebvander(x_grid, degree).T

        self.register_buffer('B', torch.from_numpy(basis).float())
        self.register_buffer('B_pinv', torch.from_numpy(pinv(basis)).float())

        # 2. Define Learnable Parameters (Low-Rank Decoupling)
        # Temporal Coefficients C: [N, R, M, degree+1]
        self.C = nn.Parameter(torch.randn(num_syn, rank, num_segs, degree+1))
        # Singular Values Sigma: [N, R]
        self.S = nn.Parameter(torch.rand(num_syn, rank))
        # Spatial Basis V: [N, channels, R]
        self.V = nn.Parameter(torch.randn(num_syn, channels, rank))

    @torch.no_grad()
    def init_params(self, x_real):
        """ Initialize via Manifold-Aware Projection """
        # x_real: [N, T, channels]

        # A. Perform SVD: X ~ U * Sigma * V^T
        U, S, Vh = torch.linalg.svd(x_real, full_matrices=False)

        # B. Truncate to Rank R (Low-Rank Constraint)
        U_k = U[..., :self.rank] # [N, T, R]
        S_k = S[..., :self.rank] # [N, R]
        Vh_k = Vh[..., :self.rank, :] # [N, R, channels]

        # C. Assign Spatial & Spectral Params
        self.S.data.copy_(S_k)
        self.V.data.copy_(Vh_k.permute(0, 2, 1)) # Store as [N, channels, R]

        # D. Project Temporal U onto Chebyshev Basis: C = U * B_pinv
        # Reshape U_k to segments: [N, R, M, seg_len]
        U_segs = U_k.view(..., self.num_segs, self.seg_len)
        self.C.data.copy_(U_segs @ self.B_pinv)

    def forward(self):
        # 1. Reconstruct Temporal Basis U: C @ B -> [N, R, M, seg_len]
        U_time = (self.C @ self.B).view(self.num_syn, self.rank, -1)

        # 2. Weight by Singular Values: U * Sigma
        U_weighted = U_time * self.S.unsqueeze(-1) # [N, R, T]

        # 3. Combine with Spatial Basis V: (U * Sigma)^T @ V^T
        # Result Shape: [N, T, channels]
        return torch.matmul(U_weighted.permute(0, 2, 1), self.V.permute(0, 2, 1))
```

*Table 5.* Statistics and specifications of the six HAR datasets.

|  | **ActR** | **MeR** | **FacT** | **NTU-Fi** | **PAMAP2** | **UCI-HAR** |
|---|---|---|---|---|---|---|
| **Sensor Modality** | WiFi (CSI) | WiFi (CSI) | WiFi (CSI) | WiFi (CSI) | Wearable (IMU) | Phone (IMU) |
| **Activity Classes** | 7 | 7 | 7 | 6 | 13 | 6 |
| **Sampling Rate** | 1000 Hz | 1000 Hz | 1000 Hz | 500 Hz | 100 Hz | 50 Hz |
| **Training Samples** | 3959 | 3959 | 3959 | 936 | 25442 | 6401 |
| **Validation Samples** | 990 | 990 | 990 | 264 | 6361 | 1601 |
| **Resolution** ($N$) | 2000 | 2000 | 2000 | 500 | 128 | 128 |
| **Sensor Channels** ($S$) | 30 | 30 | 30 | 342 | 36 | 9 |

**FacT**: A spacious environmental dataset recorded in an open factory setting with minimal obstacles. Unlike the confined spaces of the activity and meeting rooms, we selected this setting to offer a cleaner channel state with fewer multipath components. It contains nearly 5000 samples across the same 7 activity classes.

### D.2.2. PUBLIC BENCHMARKS

**NTU-Fi** (Yang et al., 2023)[4]: A large-scale WiFi-based HAR dataset collected using the Intel 5300 NIC tool. It captures CSI from 3 antennas across 30 subcarriers. The dataset involves multiple participants performing 6 distinct activities (e.g., gait, fall, gesture) in different indoor environments. It serves as a challenging benchmark for testing the robustness of distilled data against high-dimensional spectral noise and varying signal resolutions.

**PAMAP2** (Reiss & Stricker, 2012)[5]: A comprehensive physical activity monitoring dataset comprising data from 9 subjects wearing 3 Inertial Measurement Units (IMUs) on the hand, chest, and ankle. It records 18 different physical activities (such as running, cycling, and vacuuming) at a high sampling rate of 100 Hz. With over 40 channels of sensor readings (accelerometer, gyroscope, magnetometer, temperature), PAMAP2 provides a rigorous testbed for modeling complex spatial dependencies and rapid motion transients.

**UCI-HAR** (Anguita et al., 2013)[6]: A standard smartphone-based benchmark collected from 30 volunteers using a waist-mounted Samsung Galaxy S II. The dataset captures 3-axial linear acceleration and 3-axial angular velocity at a constant rate of 50 Hz. It includes 6 basic daily activities: walking, walking upstairs, walking downstairs, sitting, standing, and laying. Due to its popularity, UCI-HAR serves as a fundamental baseline to verify the generalizability of our distillation method on widely accessible consumer devices.

Furthermore, we present the data dimensions and detailed statistics for each dataset in Table 5.

### D.3. Detail of Baselines

We briefly introduce these baselines and provide the code used for their implementation.

**Random Selection.** As the most fundamental baseline, Random Selection constructs the synthetic set by uniformly sampling images from the original training dataset without replacement. While simple, it serves as a critical lower bound to verify whether sophisticated distillation methods truly learn effective representations beyond naive data subsampling.

**K-Means.** (Ahmed et al., 2020) K-Means employs a clustering-based strategy to ensure representativeness. It first partitions the training samples of each class into $k$ clusters (where $k$ equals the target IPC) in the raw pixel space. The synthetic set is then formed by selecting the centroid of each cluster or the sample nearest to the centroid, effectively capturing the geometric center of the data distribution.

**Herding.** (Welling, 2009) Herding adopts a deterministic selection approach based on moment matching. It iteratively selects samples that shift the empirical mean of the selected subset closer to the true mean of the entire class distribution. By aligning the first-order statistics, Herding ensures that the condensed set preserves the central tendency of the original information.

**K-Center.** (Sener & Savarese, 2018) K-Center, originally proposed for active learning, focuses on geometric coverage. It

---

[4]https://github.com/xyanchen/WiFi-CSI-Sensing-Benchmark
[5]https://archive.ics.uci.edu/dataset/231/pamap2+physical+activity+monitoring
[6]https://archive.ics.uci.edu/dataset/240/human+activity+recognition+using+smartphones

utilizes a greedy strategy to select a subset of samples such that the maximum distance between any data point in the full dataset and its nearest selected sample is minimized. This approach ensures that the synthetic set covers the support of the data distribution, including outliers and boundary cases.

**Dataset Condensation (DC).** [7](Zhao et al., 2021) Dataset Condensation (DC) pioneers the gradient matching paradigm. It treats the synthetic data as learnable parameters and optimizes them by matching the gradients of the network weights induced by the synthetic batch with those induced by the real batch. This bi-level optimization ensures that a model trained on synthetic data undergoes similar parameter updates as one trained on real data.

**Matching Training Trajectories (MTT).** [8](Cazenavette et al., 2022) Matching Training Trajectories (MTT) extends the short-term gradient matching of DC to long-term trajectory alignment. Instead of matching gradients at a single step, MTT unrolls the training process for multiple steps and minimizes the discrepancy between the parameters of student networks trained on real and synthetic trajectories. This results in superior performance, particularly for complex vision tasks, though with higher computational costs.

**DANCE** [9](Zhang et al., 2024). DANCE propose a dual-view approach to dataset condensation. By interpolating between initialized and trained models, our method performs pseudo long-term alignment to stabilize inner-class distributions without expensive training. Simultaneously, it uses expert models for distribution calibration, effectively mitigating inter-class shifts and maintaining class separability. We adopt DANCE as the primary optimization objective for our CHESS framework due to its high training efficiency and stability.

**NCFM.** [10](Wang et al., 2025) Neural Characteristic Function Matching (NCFM) reformulates dataset distillation as a minmax optimization problem. Unlike previous methods relying on Maximum Mean Discrepancy (MMD) which only matches moments, NCFM introduces Neural Characteristic Function Discrepancy (NCFD). By aligning both the phase and amplitude of neural features in the complex plane, it captures comprehensive distributional information, achieving a balance between realism and diversity with high computational efficiency.

## D.4. Network Architectures

In this section, we introduce the architectures used in our experiments and the experimental architectures used in cross-architecture experiments:

**ConvNet.** Following previous studies (Zhao et al., 2021; Zhang et al., 2024; Wang et al., 2025), we leverage the ConvNet as the default network architecture for both distillation and evaluation of synthetic datasets. It is a convolutional neural network designed to handle sensor data inputs. The network consists of three sequential blocks. Each block comprises a convolution layer with $3 \times 3$ filters and padding of 1, followed by a batch normalization layer, a ReLU activation function, and a max-pooling layer with a $2 \times 2$ kernel and stride of 2. The channel depths for the three blocks progress as 32, 64, and 128, respectively. Finally, a linear classifier flattens the feature map to output the logits.

**ResNet.** We adapt the standard ResNet architecture (He et al., 2016) to fit the specific input dimensions of our sensor datasets, denoted as ResNet. It is constructed using three stages of Basic Residual Blocks. Each block contains two $3 \times 3$ convolutions with batch normalization and ReLU activation, accompanied by a shortcut connection to mitigate gradient vanishing. The network utilizes max-pooling between stages for downsampling and concludes with a fully connected layer. We utilize this network to evaluate the cross-architecture generalization of our proposed method.

**MobileNet.** To evaluate performance on lightweight architectures, we employ MobileNet (Howard et al., 2017). This model integrates the core Depthwise Separable Convolution blocks to reduce computational cost. It is structured with an initial standard convolution layer followed by two stages of depthwise separable convolutions. Each stage increases the channel width up to 128 while reducing spatial resolution via max-pooling. This design allows us to verify the efficacy of synthetic data on parameter-efficient models.

**ShuffleNet.** ShuffleNet (Zhang et al., 2018) incorporates channel shuffle operations and grouped convolutions to facilitate efficient information flow between channel groups. The architecture comprises a standard initial convolution followed by two ShuffleNet units with group sizes of 4 and 8, respectively. Similar to the other adapted models, it uses max-pooling for downsampling. This structure serves as a robust metric for evaluating performance on architectures with sparse connections.

---

[7] https://github.com/VICO-UoE/DatasetCondensation
[8] https://github.com/georgecazenavette/mtt-distillation
[9] https://github.com/Hansong-Zhang/DANCE
[10] https://github.com/gszfwsb/NCFM

**AlexNet.** We adapt the classic AlexNet structure (Krizhevsky et al., 2012) by tailoring the kernel sizes to accommodate the non-square shape of sensor inputs. Specifically, the first two layers utilize rectangular kernels of sizes $7 \times 3$ and $5 \times 3$ with asymmetric padding, while the third layer uses a standard $3 \times 3$ kernel. Each convolution is followed by batch normalization, ReLU, and max-pooling. This adapted version, AlexNet, allows us to assess the effectiveness of the synthetic data on legacy-style architectures with large receptive fields.

**THAT.**[11] THAT (Li et al., 2021) adopt the Two-stream Convolution Augmented Transformer to capture long-range temporal dependencies and inter-channel correlations. This model employs a two-tower structure that processes temporal and channel streams independently through Multi-scale Convolution Augmented Transformer (MCAT) layers. Specifically, the MCAT integrates multi-head self-attention with multi-scale convolutional blocks, utilizing kernel sizes of $\{1, 3, 5\}$ for the temporal module and $\{1, 2, 3\}$ for the channel module to extract range-based patterns. Furthermore, a Gaussian range encoding mechanism is incorporated to preserve order-sensitive positional information. Including THAT allows us to verify the generalizability of our synthesized data on sophisticated attention-based backbones.

### D.5. Hyperparameter Setting

The hyperparameters of baselines and CHESS are listed in Tables 6 and 7, respectively

*Table 6.* Hyperparameters used for Data Synthesis across six datasets.

|  | **ActR** | **MeR** | **FacT** | **NTU-Fi** | **PAMAP2** | **UCI-HAR** |
|---|---|---|---|---|---|---|
| Optimizer | Adam | Adam | Adam | Adam | Adam | Adam |
| Initial LR | 0.000001 | 0.000001 | 0.000001 | 0.00001 | 0.00001 | 0.00001 |
| Batch Size | 256 | 256 | 256 | 256 | 256 | 256 |
| Iterations | 50000 | 50000 | 50000 | 50000 | 50000 | 50000 |
| Rank ($R$) | 5 | 5 | 5 | 50 | 8 | 3 |
| Segment Length ($L$) | 100 | 100 | 100 | 5 | 8 | 8 |
| Degree ($D$) | 2 | 2 | 2 | 2 | 2 | 2 |
| Expert Model Number | 20 | 20 | 20 | 20 | 20 | 20 |

*Table 7.* Hyperparameters used for Validation.

|  | **ActR** | **MeR** | **FacT** | **NTU-Fi** | **PAMAP2** | **UCI-HAR** |
|---|---|---|---|---|---|---|
| Optimizer | Adam | Adam | Adam | Adam | Adam | Adam |
| Initial LR | 0.0001 | 0.0001 | 0.0001 | 0.0005 | 0.001 | 0.001 |
| Batch Size | 256 | 256 | 256 | 256 | 256 | 256 |
| Epochs | 300 | 300 | 300 | 500 | 2000 | 500 |

### D.6. Performance Comparison under Standardized BPC Settings

Following the standardized Budget Per Class (BPC) evaluation framework established in the previous DD parameterization work (Kim et al., 2022b; Wei et al., 2023; Shin et al.), we assess our method's synthesis efficiency under a strictly constrained budget of BPC=1. As shown in Table 8, our CHESS strategy significantly outperforms the DANCE baseline across all six datasets, demonstrating much higher information density per parameter. Notably, while maintaining the same storage footprint, CHESS achieves substantial relative accuracy gains ranging from 37.0% to 135.9%. These results validate that our structured parameterization is far more effective at capturing the essential dynamics of CTDS signals than the unconstrained pixel-wise optimization.

### D.7. More Cross-Architecture Generalization Results

In this section, we analyze four additional datasets including ActR (SPC=5), FacT (SPC=5), NTU-FI (SPC=1) and PAMAP2 (SPC=5) as shown in Table 9. These diverse sensor types further reveal the weakness of conventional pixel-wise DD

---

[11] https://github.com/windofshadow/THAT

*Table 8.* Performance Comparison and Accuracy Improvement at Fixed BPC=1.

|  | ActR | MeR | FacT | NTU-Fi | PAMAP2 | UCI-HAR |
|---|---|---|---|---|---|---|
| *DANCE* | $40.1 \pm 1.6$ | $47.2 \pm 1.4$ | $33.1 \pm 1.7$ | $68.9 \pm 2.1$ | $45.2 \pm 0.9$ | $59.4 \pm 1.2$ |
| **CHESS (Ours)** | $81.6 \pm 2.2$ | $85.3 \pm 2.3$ | $78.1 \pm 1.4$ | $94.4 \pm 4.5$ | $66.6 \pm 0.9$ | $83.2 \pm 0.5$ |
|  | (+103.5%) | (+80.7%) | (+135.9%) | (+37.0%) | (+47.3%) | (+40.1%) |
| Sample (IPC) | 133 | 133 | 133 | 5 | 6 | 6 |

*Table 9.* Cross-architecture performance comparison on ActR (SPC=5), FacT (SPC=5), NTU-FI (SPC=1) and PAMAP2 datasets (SPC=5). Highlights indicate the best , second-best , and third-best results.

| Method | ResNet | AlexNet | ShuffleNet | MobileNet | THAT |
|---|---|---|---|---|---|
| **ActR** | | | | | |
| DC | $61.8_{\pm 3.5}$ | $59.4_{\pm 3.6}$ | $59.3_{\pm 3.6}$ | $65.5_{\pm 3.9}$ | $54.9_{\pm 1.5}$ |
| MTT | $50.8_{\pm 1.9}$ | $49.1_{\pm 2.4}$ | $40.7_{\pm 6.6}$ | $49.5_{\pm 3.6}$ | $55.1_{\pm 0.3}$ |
| DANCE | $43.2_{\pm 3.1}$ | $49.1_{\pm 3.0}$ | $38.3_{\pm 4.4}$ | $44.6_{\pm 8.2}$ | $56.3_{\pm 1.9}$ |
| NCFM | $55.0_{\pm 5.8}$ | $61.3_{\pm 2.9}$ | $51.7_{\pm 6.6}$ | $57.4_{\pm 3.6}$ | $55.5_{\pm 1.4}$ |
| CHESS w/o SVD | $64.5_{\pm 1.2}$ | $68.8_{\pm 1.3}$ | $52.5_{\pm 6.2}$ | $63.5_{\pm 1.7}$ | $55.2_{\pm 1.0}$ |
| CHESS w/o C.P | $52.3_{\pm 1.6}$ | $62.1_{\pm 1.7}$ | $43.2_{\pm 2.1}$ | $51.2_{\pm 1.6}$ | $55.7_{\pm 2.3}$ |
| **CHESS** | $\mathbf{65.0}_{\pm 2.7}$ | $\mathbf{70.6}_{\pm 1.8}$ | $\mathbf{59.6}_{\pm 5.1}$ | $\mathbf{67.1}_{\pm 4.3}$ | $\mathbf{57.7}_{\pm 0.2}$ |
| **FacT** | | | | | |
| DC | $55.0_{\pm 2.7}$ | $60.5_{\pm 2.1}$ | $47.7_{\pm 7.8}$ | $53.2_{\pm 2.6}$ | $45.2_{\pm 2.8}$ |
| MTT | $44.5_{\pm 1.1}$ | $48.9_{\pm 1.1}$ | $44.4_{\pm 3.6}$ | $44.8_{\pm 1.3}$ | $46.1_{\pm 0.6}$ |
| DANCE | $38.1_{\pm 4.2}$ | $47.2_{\pm 4.1}$ | $29.9_{\pm 2.9}$ | $33.1_{\pm 4.6}$ | $45.6_{\pm 0.8}$ |
| NCFM | $44.0_{\pm 4.6}$ | $48.8_{\pm 3.7}$ | $39.4_{\pm 10.3}$ | $41.7_{\pm 3.7}$ | $46.7_{\pm 1.3}$ |
| CHESS w/o SVD | $60.4_{\pm 2.1}$ | $63.3_{\pm 0.8}$ | $50.8_{\pm 0.8}$ | $53.4_{\pm 1.2}$ | $46.3_{\pm 3.8}$ |
| CHESS w/o C.P | $52.4_{\pm 3.5}$ | $57.1_{\pm 2.3}$ | $49.4_{\pm 0.9}$ | $49.4_{\pm 2.5}$ | $45.1_{\pm 1.5}$ |
| **CHESS** | $\mathbf{61.6}_{\pm 1.0}$ | $\mathbf{63.9}_{\pm 1.7}$ | $\mathbf{51.1}_{\pm 5.9}$ | $\mathbf{56.9}_{\pm 5.9}$ | $\mathbf{46.9}_{\pm 1.8}$ |
| **NTU-FI** | | | | | |
| DC | $46.8_{\pm 8.5}$ | $52.4_{\pm 3.2}$ | $42.2_{\pm 13.7}$ | $41.8_{\pm 7.2}$ | $32.2_{\pm 1.9}$ |
| MTT | $35.9_{\pm 3.3}$ | $45.8_{\pm 0.9}$ | $39.8_{\pm 11.3}$ | $34.4_{\pm 8.5}$ | $32.7_{\pm 8.1}$ |
| DANCE | $40.6_{\pm 8.6}$ | $48.8_{\pm 12.0}$ | $34.1_{\pm 5.5}$ | $39.6_{\pm 4.2}$ | $\mathbf{33.1}_{\pm 12.6}$ |
| NCFM | $47.1_{\pm 6.1}$ | $48.8_{\pm 12.0}$ | $30.6_{\pm 6.9}$ | $39.3_{\pm 4.6}$ | $28.9_{\pm 14.5}$ |
| CHESS w/o SVD | $54.5_{\pm 2.4}$ | $57.8_{\pm 2.1}$ | $46.0_{\pm 4.3}$ | $42.0_{\pm 1.3}$ | $32.1_{\pm 2.7}$ |
| CHESS w/o C.P | $50.2_{\pm 2.3}$ | $49.1_{\pm 3.5}$ | $41.2_{\pm 5.2}$ | $40.2_{\pm 5.6}$ | $31.7_{\pm 0.9}$ |
| **CHESS** | $\mathbf{54.7}_{\pm 2.1}$ | $\mathbf{58.5}_{\pm 6.5}$ | $\mathbf{46.3}_{\pm 7.0}$ | $\mathbf{42.7}_{\pm 7.4}$ | $32.8_{\pm 4.3}$ |
| **PAMAP2** | | | | | |
| DC | $60.8_{\pm 2.3}$ | $64.7_{\pm 1.2}$ | $53.7_{\pm 4.3}$ | $59.4_{\pm 1.7}$ | $55.0_{\pm 4.4}$ |
| MTT | $48.5_{\pm 1.9}$ | $51.4_{\pm 0.7}$ | $43.5_{\pm 3.7}$ | $47.8_{\pm 1.6}$ | $54.3_{\pm 1.1}$ |
| DANCE | $47.3_{\pm 2.0}$ | $55.7_{\pm 0.8}$ | $43.8_{\pm 3.3}$ | $50.5_{\pm 1.5}$ | $53.4_{\pm 1.8}$ |
| NCFM | $50.8_{\pm 2.3}$ | $59.2_{\pm 0.6}$ | $45.6_{\pm 4.1}$ | $52.0_{\pm 2.3}$ | $53.5_{\pm 2.3}$ |
| CHESS w/o SVD | $59.4_{\pm 1.4}$ | $64.3_{\pm 2.1}$ | $\mathbf{55.1}_{\pm 1.3}$ | $58.8_{\pm 2.6}$ | $54.5_{\pm 1.9}$ |
| CHESS w/o C.P | $55.8_{\pm 3.2}$ | $59.1_{\pm 5.2}$ | $50.2_{\pm 1.9}$ | $54.9_{\pm 2.9}$ | $49.2_{\pm 0.7}$ |
| **CHESS** | $\mathbf{61.2}_{\pm 0.2}$ | $\mathbf{65.5}_{\pm 0.5}$ | $55.0_{\pm 1.7}$ | $\mathbf{59.8}_{\pm 1.5}$ | $\mathbf{55.7}_{\pm 0.1}$ |

methods on lightweight networks. For example, on the FacT dataset, DANCE fails on ShuffleNet and achieves only 29.9% accuracy, whereas CHESS maintains a stable score of 51.1%. Similarly, on the NTU-FI dataset, CHESS beats the best baseline by about 4% on ShuffleNet, proving it works well even with complex WiFi signals. Furthermore, regarding the Transformer-based architecture THAT, CHESS demonstrates superior generalization capabilities compared to DANCE. On the PAMAP2 dataset, CHESS achieves 55.7%, outperforming DANCE which reaches 53.4% by a margin of 2.3%. Similarly, on the FacT dataset, CHESS with 46.9% accuracy surpasses DANCE at 45.6% by 1.3%. This indicates that our manifold-constrained synthesis effectively preserves the long-range temporal dependencies required by self-attention mechanisms, which are often lost by pixel-wise optimization methods. On ActR, while baselines like MTT struggle on ShuffleNet with 40.7% accuracy, CHESS secures a strong 59.6%. The ablation variants provide extra proof for our design in these new cases. Removing the Chebyshev Polynomials constraint, denoted as *w/o C.P*, causes a large performance drop on ActR with ResNet from 65.0% down to 52.3%. This shows that temporal smoothness is needed for generalization. Likewise, the variant without the low-rank constraint, denoted as *w/o SVD*, underperforms the full model on ActR with ShuffleNet by 7.1%. This confirms that spatial structural priors are key to keeping features useful across different sensor modalities.

### D.8. More Zero-Shot Resolution Adaptation Results

To rigorously assess the impact of our resolution-agnostic decoding mechanism, we conduct a comprehensive evaluation that includes both an internal ablation study and a comparative analysis against standard baseline strategies DANCE (Zhang et al., 2024) and NCFM (Wang et al., 2025). All models are distilled at a reference resolution of $L = 1000$ and subsequently adapted to unseen target resolutions of $L = 200$ and $L = 2000$. For baseline comparisons, we employ standard linear pooling for downsampling and discrete interpolation strategies including Nearest, Linear, and Cubic for upsampling. Crucially, to isolate the efficacy of our proposed decoding, we construct internal ablation variants where these identical discrete downsampling and interpolation techniques are applied directly to the complete tensors distilled by CHESS. We then contrast these discrete variants with our proposed Analytical Resampling which leverages the continuous manifold properties of the synthesized data.

We conduct experiments on three high-sampling-rate WiFi CSI datasets we collected:ActR, MeR, and FacT with a budget of SPC=5. As detailed in Table 10, Analytical Resampling consistently yields superior performance across all datasets and target resolutions. This advantage is particularly pronounced in the complex FacT and ActR datasets. For instance, on the FacT downsampling task ($L = 200$), Analytical Resampling achieves 66.6% accuracy, outperforming the linear interpolation baseline of 64.9% by a significant margin of 1.7%. Similarly, in the upsampling regime ($L = 2000$) for ActR, our analytical approach surpasses linear interpolation by 1.6%, improving accuracy from 69.8% to 71.4%. By mathematically re-evaluating the Chebyshev basis matrix on the new target grid, CHESS bypasses these lossy approximations, preserving signal details and ensuring global coherence regardless of the resolution shift.

### D.9. Hyperparameter Analysis

In this section, we conduct a comprehensive sensitivity analysis to evaluate the impact of the three core hyperparameters governing our strategy: the rank $R$, the segment length $L$, and the polynomial degree $D$. We perform these experiments across six diverse datasets to verify the robustness and generalization capability of the proposed method. Our primary objective is to elucidate the trade-off mechanisms between the compression ratio $\gamma$ and classification accuracy. By systematically varying each parameter while keeping others fixed, we identify the optimal operating boundaries that maximize storage efficiency without compromising the semantic fidelity of the synthesized sensor data. The following analysis details the individual and joint effects of these parameters on system performance.

**Effect of Low-Rank Constraint $R$.** As displayed in the first column of Fig. 5, we first examine the impact of the rank $R$ across all datasets. We observe a sharp increase in accuracy as $R$ rises from minimal values, which confirms that a small subspace is sufficient to capture the signal manifold. However, performance saturates beyond a certain threshold, indicating that the intrinsic dimensionality of the sensor data is indeed low. For instance, the accuracy on ActR and MeR datasets stabilizes when $R$ reaches 5, whereas NTU-Fi require a slightly higher rank around 10 to capture their richer signal variations due to its samples have more channels. Notably, further increasing $R$ does not yield gains but rather risks overfitting to architecture-specific noise. Since increasing $R$ also linearly expands the storage budget and causes the compression ratio $\gamma$ to drop significantly, we select the saturation point as the optimal operating point as shown in Table D.5. This choice maximizes parameter efficiency while fully preserving the semantic information of the latent channels.

On the other hand, CHESS exhibits controlled sensitivity to violations of the low-rank assumption. When this assumption is

*Table 10.* Zero-shot resolution adaptation performance. Models are distilled at $L = 1000$. For baselines, downsampling ($L = 200$) uses standard linear pooling, while upsampling ($L = 2000$) relies on interpolation strategies (Nearest, Linear, Cubic). To verify the effectiveness of our Analytical Resampling, we explicitly compare it against standard discrete resampling strategies (Nearest, Linear, Cubic) applied to the complete sample distilled by CHESS. Highlights indicate the **best** results.

| Method | Variant / Mode | Target Resolution ($L'$) | | |
|---|---|---|---|---|
| | | 200 (Down) | 1000 (Train) | 2000 (Up) |
| *ActR* | | | | |
| | Nearest | | | $60.6 \pm 0.7$ |
| *DANCE* | Linear | $66.1 \pm 1.8$ | $63.5 \pm 2.8$ | $61.9 \pm 0.9$ |
| | Cubic | | | $60.8 \pm 0.4$ |
| | Nearest | | | $62.6 \pm 1.2$ |
| *NCFM* | Linear | $65.6 \pm 1.2$ | $67.9 \pm 1.5$ | $65.3 \pm 1.1$ |
| | Cubic | | | $64.1 \pm 1.8$ |
| | Nearest | | | $70.4 \pm 0.2$ |
| | Linear | $75.8 \pm 0.6$ | | $69.8 \pm 0.1$ |
| *CHESS* | Cubic | | | $68.5 \pm 0.4$ |
| | Analytical Resampling | $\mathbf{76.6 \pm 0.4}$ | $\mathbf{72.3 \pm 0.2}$ | $\mathbf{71.4 \pm 0.8}$ |
| *MeR* | | | | |
| | Nearest | | | $64.3 \pm 1.2$ |
| *DANCE* | Linear | $66.3 \pm 0.7$ | $65.2 \pm 0.1$ | $62.3 \pm 0.9$ |
| | Cubic | | | $65.6 \pm 1.1$ |
| | Nearest | | | $64.2 \pm 2.1$ |
| *NCFM* | Linear | $64.2 \pm 0.8$ | $63.4 \pm 1.0$ | $62.1 \pm 2.0$ |
| | Cubic | | | $62.8 \pm 0.8$ |
| | Nearest | | | $73.5 \pm 0.4$ |
| | Linear | $76.4 \pm 0.7$ | | $74.6 \pm 0.7$ |
| *CHESS* | Cubic | | | $74.5 \pm 0.2$ |
| | Analytical Resampling | $\mathbf{76.9 \pm 0.2}$ | $\mathbf{76.2 \pm 0.2}$ | $\mathbf{75.8 \pm 1.1}$ |
| *FacT* | | | | |
| | Nearest | | | $45.2 \pm 1.0$ |
| *DANCE* | Linear | $41.2 \pm 0.9$ | $38.6 \pm 0.7$ | $53.2 \pm 1.1$ |
| | Cubic | | | $47.8 \pm 0.2$ |
| | Nearest | | | $44.2 \pm 1.1$ |
| *NCFM* | Linear | $37.1 \pm 1.2$ | $35.4 \pm 0.9$ | $42.4 \pm 0.2$ |
| | Cubic | | | $47.0 \pm 0.3$ |
| | Nearest | | | $64.7 \pm 0.2$ |
| | Linear | $64.9 \pm 0.3$ | | $63.8 \pm 0.3$ |
| *CHESS* | Cubic | | | $63.9 \pm 0.4$ |
| | Analytical Resampling | $\mathbf{66.6 \pm 0.2}$ | $\mathbf{65.7 \pm 0.5}$ | $\mathbf{65.4 \pm 0.8}$ |

strongly violated (e.g., independent multivariate signals with high intrinsic dimensionality), enforcing a small $R$ induces underfitting. However, the degradation is gradual rather than catastrophic. As shown in Fig. 5, even under severe rank mismatch (e.g., $R = 1, 2$), performance declines gracefully ($\sim 10\%$ absolute drop) without collapse.

**Impact of Temporal Segmentation Length** $L$. The second column of Fig. 5 reveals a critical trade-off regarding the segment length $L$. Increasing $L$ substantially boosts the compression ratio $\gamma$ by reducing the total number of required coefficients. Nevertheless, the accuracy trends exhibit distinct behaviors across different modalities. For high sampling rate WIFI CSI datasets like ActR and MeR, the performance remains robust even at larger segment lengths. This stability implies

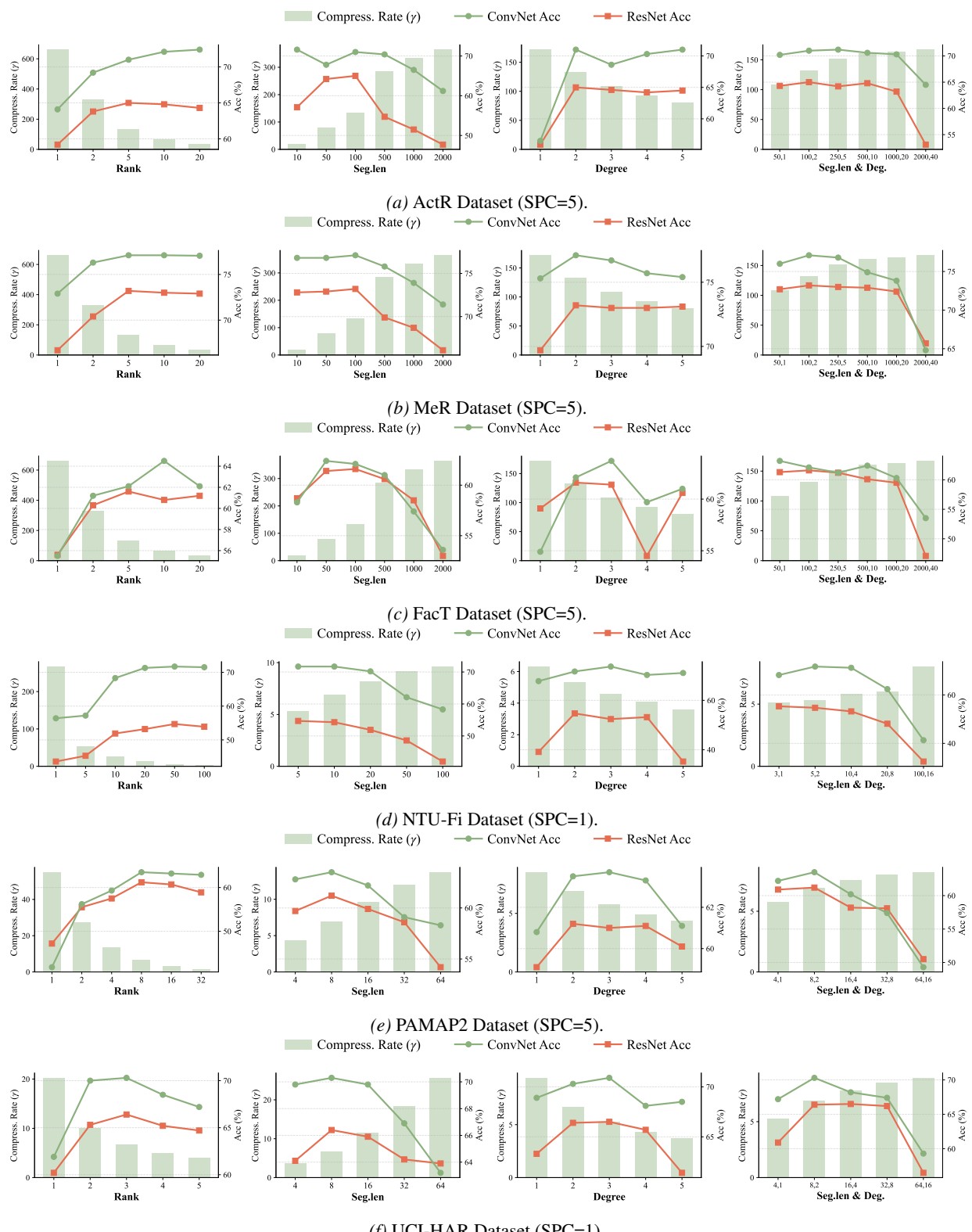

*Figure 5.* Hyperparameter sensitivity analysis across six datasets. We evaluate the impact of rank $R$, segment length $L$, and polynomial degree $D$ on both the compression ratio $\gamma$ (bars, left axis) and classification accuracy (lines, right axis). The rightmost plot analyzes the parameter allocation trade-off (segment length $L$, and polynomial degree $D$) under high compression regimes.

that these signals exhibit stationarity over wider temporal windows. In contrast, for wearable sensor datasets including PAMAP2 and UCI-HAR, the accuracy deteriorates sharply when $L$ exceeds a specific limit, such as 32. This decline indicates that these signals contain high-frequency dynamics where excessively long segments fail to capture rapid temporal variations due to underfitting. Therefore, the optimal $L$ is dataset-dependent and is determined by the sampling rate.

**Influence of Polynomial Degree $D$.** The third column of Fig. 5 illustrates the impact of the polynomial degree $D$. Across all datasets, the results demonstrate that a low degree is generally sufficient to achieve near-optimal accuracy. Moving from a linear to a quadratic approximation provides a significant accuracy boost by capturing the non-linear curvature of the signal. However, increasing $D$ further leads to performance saturation or even degradation as seen in the FacT and PAMAP2 cases. This drop can be attributed to the Runge phenomenon (Runge et al., 1901) where high-degree polynomials introduce spurious oscillations between data points. Since higher degrees also consume more storage budget, a small degree such as 2 is empirically identified as the most efficient choice.

**Parameter Allocation Strategy.** Finally, the rightmost column of Figure 5 investigates the optimal parameter allocation strategy under a constrained storage budget. This analysis explores the critical trade-off between temporal granularity (segment length $L$) and functional expressiveness (polynomial degree $D$) by contrasting fine-grained low-order models against coarse high-order approximations. Across all datasets, we observe a consistent trend where configurations favoring local piecewise smoothness (specifically $L = 100, D = 2$ for WiFi datasets and $L = 8, D = 2$ for IMU datasets) significantly outperform global high-order fitting strategies. For instance, on the ActR dataset, shifting towards a coarse-grained setting with $L = 2000$ and $D = 40$ leads to a catastrophic performance drop, despite the high theoretical capacity of degree-40 polynomials. This degradation is primarily attributed to Runge's Phenomenon (Runge et al., 1901), where high-degree polynomials exhibit spurious oscillations at interval boundaries, introducing non-physical artifacts that disrupt the learning process. Consequently, we conclude that allocating the limited parameter budget towards more segments with lower degrees yields superior fidelity and generalization compared to increasing the polynomial order. Moreover, this piecewise segmentation strategy plays a key role in making the theoretical derivative bound (as shown in Proposition 1 and Lemma 1) effective in practice. If a single global polynomial were employed for the entire sequence, the required degree $D$ would scale linearly with sequence length, causing the derivative bound $D^2$ to explode and lose its constraining power. It is precisely this piecewise low-degree parameterization that substantiates the derivative bound, ensuring it remains a tight and physically meaningful constraint rather than a loose theoretical upper limit.

Additional discussions regarding hyperparameter selection in practice are provided in Appendix E.4.

### D.10. Computational Efficiency Analysis

*Table 11.* Comparison of computational efficiency. We report the average training time (per 50 iterations) and peak GPU memory consumption. CHESS achieves a reduction in training time due to the compact parameter space, with only negligible memory overhead for graph construction.

| Method | MeR | | UCI-HAR | |
|---|---|---|---|---|
| | Time (s) $\downarrow$ | Memory (MB) $\downarrow$ | Time (s) $\downarrow$ | Memory (MB) $\downarrow$ |
| *DANCE* | 15.31 | 21104.6 | 3.94 | 2150.0 |
| **CHESS** | **14.55** (-5.0%) | 21176.3 (+0.3%) | **3.85** (-2.3%) | 2198.0 (+2.2%) |

We quantitatively evaluate the computational cost of CHESS by recording the average training duration per 50 iterations and peak GPU memory usage on MeR dataset and UCI-HAR dataset, as detailed in Table 11. Contrary to the intuition that constructing continuous polynomial trajectories adds computational burden, results show that CHESS actually accelerates the training process. Specifically, the method reduces latency by 5.0% on MeR and 2.3% on UCI-HAR. This efficiency gain stems from the low-rank parameterization: optimization is performed on a compact set of the spatial basis $\mathbf{V}_{space}$, the set of temporal coefficient matrices $\{\mathbf{C}_m\}_{m=1}^M$, and the singular values $\boldsymbol{\sigma}$ rather than the high-dimensional raw tensor space, leading to faster convergence per step. Regarding memory, the explicit construction of the computational graph for trajectory reconstruction in Eq. 8 introduces only a negligible overhead ranging from 0.3% to 2.2%. This demonstrates that CHESS achieves superior storage compression and continuous modeling capabilities without compromising training efficiency.

# E. More Discussion

## E.1. Is Boundary Constraint Necessary in Our Work?

In principle, our goal is to parameterize the alignment target as a globally continuous function to reflect the underlying physical dynamics. Ideally, a single high-degree polynomial could serve this purpose. However, such global approximations are notoriously unstable on uniform grids due to Runge's phenomenon (Runge et al., 1901), which introduces unrealistic high-amplitude oscillations. Consequently, we are forced to depart from a single global formulation and instead employ a segment-wise approximation, which balances the need for continuity with numerical stability (Zhang et al., 2025; Zhu & Wu, 2022).

A critical design choice in CHESS lies in the treatment of segment boundaries. Intuitively, one might hypothesize that enforcing strict analytical continuity at the interfaces of adjacent segments, such as $C^0$ value continuity or $C^1$ derivative continuity, would yield superior performance by assuring physical meaningful. To empirically verify this hypothesis, we conducted a rigorous ablation study comparing our proposed *Unconstrained* optimization against two *Hard Constraint* strategies: the $C^0$ constraint which strictly enforces positional connectivity, and the $C^0 + C^1$ constraint which enforces both positional and derivative continuity as inspired by classical spline interpolation theory (De Boor & De Boor, 1978).

As shown in Table 12, contrary to physical intuition, imposing these hard constraints results in a significant degradation of classification accuracy. The $C^0 + C^1$ setting causes a particularly catastrophic drop in performance. We attribute this counter-intuitive result to three theoretical and empirical factors outlined below.

*Table 12.* Ablation study on boundary constraints on ActR datasets with SPC=5. Even when increasing the polynomial degree to $D = 4$ to compensate for the degrees of freedom lost to constraints, the strict methods fail to improve. This indicates that optimization instability, rather than parameter capacity, is the bottleneck.

| Optimization Strategy | $D = 2$ | $D = 4$ |
|---|---|---|
| Hard Constraints ($C^0 + C^1$) | $14.2_{\pm 0.1}$ | $14.3_{\pm 0.1}$ |
| Hard Constraints ($C^0$) | $58.5_{\pm 1.3}$ | $57.2_{\pm 0.8}$ |
| **Unconstrained (Ours)** | $71.0_{\pm 1.0}$ | $70.3_{\pm 1.2}$ |

**Optimization Instability.** Hard constraints inevitably consume learnable Degrees of Freedom (DoF) to satisfy boundary equations. One might argue that the performance drop in constrained models with $D = 2$ is merely due to the reduced number of free parameters. To rule out this possibility and ensure a fair comparison of model capacity regarding DoF unification, we conducted experiments utilizing higher-order polynomials where $D = 4$ for the constrained settings. Theoretically, increasing $D$ compensates for the parameters consumed by the constraints. However, as evidenced in Table 12, this increase yielded negligible improvement. For instance, the accuracy for the $C^0 + C^1$ setting only shifted from 14.2% to 14.3% equal to a random selection. This result confirms that the failure is not due to a lack of parameter capacity, but rather the intrinsic optimization instability. The strict coupling transforms the problem into a rigid dependency chain, preventing the model from fitting high-frequency changes even when provided with additional polynomial terms.

**Sequential Error Propagation.** Enforcing continuity couples all segments into a global dependency chain. As noted in sequence generation tasks (Bengio et al., 2015), this induces a domino effect where a fitting error in an early segment strictly biases the initialization of the subsequent segment. The failure of the setting with $D = 4$ further validates this observation, as increasing complexity does not mitigate the error accumulation inherent in the sequential dependency. In contrast, unconstrained approach decouples the problem into independent local regression tasks, ensuring that local outliers remain isolated.

**Robustness to Sparse Discontinuities.** Deep learning models process signal data as discrete sampled sequences, not continuous analytical functions. From a signal processing perspective, the discontinuities at the segment boundaries in our unconstrained model show merely as sparse step functions at $M - 1$ points. Neural networks (especially CNNs with pooling layers) are inherently robust to such measure-zero local variations (Boureau et al., 2010; Hornik, 1991). Therefore, strict mathematical continuity yields diminishing returns for downstream classification while imposing severe optimization costs.

Consequently, we adopt the unconstrained strategy. The Piecewise Chebyshev parameterization inherently provides a sufficient structural prior for smoothness within segments, while the removal of boundary constraints grants the necessary flexibility to capture discriminative patterns.

## E.2. Why Chebyshev Polynomials?

The choice of parameterizing temporal trajectories using Chebyshev polynomials is grounded in two fundamental theoretical motivations that directly address the generalization challenges.

**Theoretical Foundation via Stone-Weierstrass Theorem.** Our departure from discrete pixel optimization to continuous trajectory modeling is theoretically underpinned by the *Stone-Weierstrass Theorem* (Rudin, 1976), as shown in Section 2.3. The theorem states that every continuous function defined on a closed interval can be uniformly approximated as closely as desired by a polynomial function. Since real-world continuous-time signals originate from underlying physical processes, they are inherently continuous. Therefore, polynomials provide a sufficient and theoretically guaranteed hypothesis space to recover the true latent dynamics hidden within the discrete observations, justifying our polynomial parameterization strategy.

**Spike Suppression via Minimax Property.** While the Stone-Weierstrass theorem justifies the use of polynomials, it does not dictate *which* polynomial basis to use. As shown in Fig.1, our empirical observation indicates that synthetic data often suffers from high-frequency 'spikes': sudden, high-amplitude oscillations that degrade cross-model generalization.

We employ Chebyshev polynomials specifically because they possess the unique *Minimax Property*: among all monic polynomials of a given degree $D$, the (scaled) Chebyshev polynomial has the smallest maximum absolute value on the interval $[-1, 1]$. Formally, let $\mathcal{P}_D$ be the set of all monic polynomials of degree $D$. For any $P(t) \in \mathcal{P}_D$, the Chebyshev polynomial $T_D(t)$ satisfies:

$$\max_{t \in [-1,1]} \left| \frac{1}{2^{D-1}} T_D(t) \right| \leq \max_{t \in [-1,1]} |P(t)|. \tag{26}$$

This property implies that Chebyshev polynomials are the 'flattest' basis functions with the minimum uniform norm. Crucially, this amplitude minimization acts as an implicit derivative regularizer. According to Markov's inequality (Proposition 1), the upper bound of a polynomial's derivative is proportional to its maximum amplitude, i.e., $\|u'\|_\infty \propto D^2 \|u\|_\infty$. By utilizing the Chebyshev basis, we explicitly minimize the amplitude factor $\|u\|_\infty$ in this inequality. Unlike other bases that are prone to Runge's phenomenon (Runge et al., 1901), our approach mathematically constrains the worst-case deviation, thereby strictly suppressing the formation of unrealistic spikes and enforcing temporal smoothness across the distilled data.

## E.3. Empirical Verification of Smoothness and Physical Meaningful.

To rigorously assess the physical validity of the synthesized data, we analyze the first-order derivative distributions $\frac{dx}{dt}$ of the first sensor channel on both the MeR and UCI-HAR datasets as illustrated in Fig. 6 and Fig. 7. A consistent failure mode is observed in the baseline NCFM (Wang et al., 2025) where discrete pixel-wise optimization leads to severe high-frequency oscillations regardless of sequence length. On the MeR dataset depicted in Fig. 6, the baseline produces a chaotic and heavy-tailed derivative distribution with magnitudes reaching approximately $\pm 20$ which indicates severe non-physical artifacts compared to the smooth natural signals. This phenomenon is further quantitatively evidenced in the UCI-HAR results displayed in Fig. 7. Here, the derivative magnitudes of the baseline reach approximately $\pm 1.0$ which is nearly $50 \times$ larger than the natural sensor noise of roughly $\pm 0.02$ observed in the original signals. In sharp contrast, CHESS acts as a physics-informed regularization. By constraining synthesis to a piecewise polynomial manifold, it enforces intrinsic temporal continuity to yield trajectories with sharp and zero-centered derivative distributions. This observation aligns with the fundamental *Minimum Jerk Principle* in motor control theory (Flash & Hogan, 1985) which posits that physical systems inherently minimize high-order derivatives due to inertia constraints. Consequently, our method successfully eliminates the non-physical spikes observed in baselines and recovers the semantic fidelity of the underlying dynamic processes.

## E.4. How to Select Hyperparameter in Practice?

In our proposed framework, the polynomial degree $D$ and the segment length $L$ are intrinsically coupled hyperparameters that jointly determine the approximation fidelity and storage efficiency of the distilled samples. Through extensive empirical analysis, we identify that a low-degree setting, specifically $D = 2$ is sufficient to capture the local dynamics without introducing the high-frequency oscillations associated with Runge's phenomenon (Runge et al., 1901). By fixing the approximation to a piecewise quadratic form, we shift the adaptation burden to the segment length $L$, which is tuned inversely proportional to the information density of the source data. We observe that $L$ is highly sensitive to the signal sampling rate: for high-sampling-rate datasets (e.g., MeR, FacT), where the signal exhibits high local smoothness and temporal redundancy, a larger $L$ is permissible to maximize compression; conversely, for low-sampling-rate scenarios where inter-sample variation is significant, $L$ is reduced to ensure that the piecewise quadratic segments can accurately track rapid

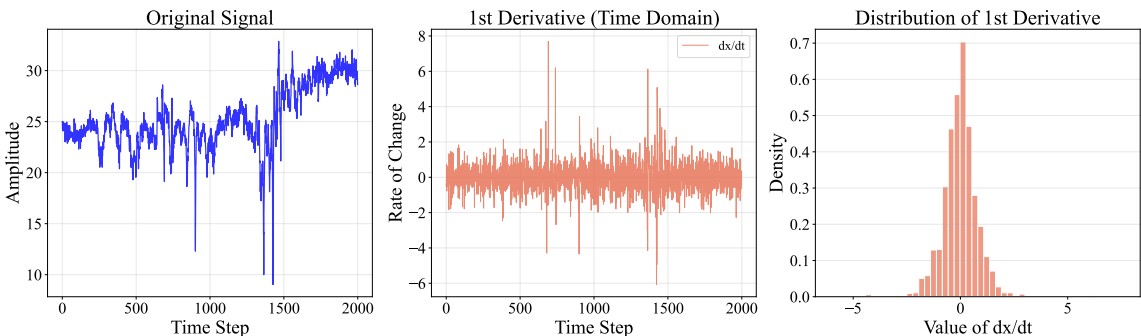

*(a)* Original signal in MeR dataset. The original sensor signal exhibits natural physical continuity.

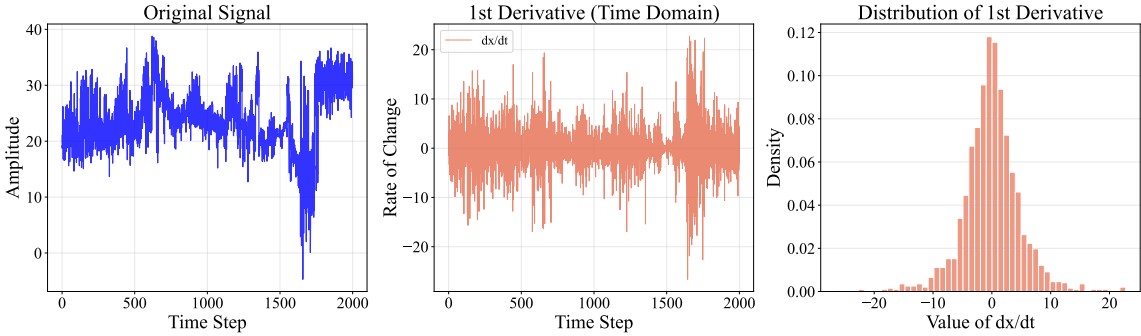

*(b)* The baseline NCFM, relying on discrete pixel-wise optimization, produces erratic fluctuations with a wide, heavy-tailed derivative distribution (range $\approx \pm 20$), indicating severe high-frequency artifacts.

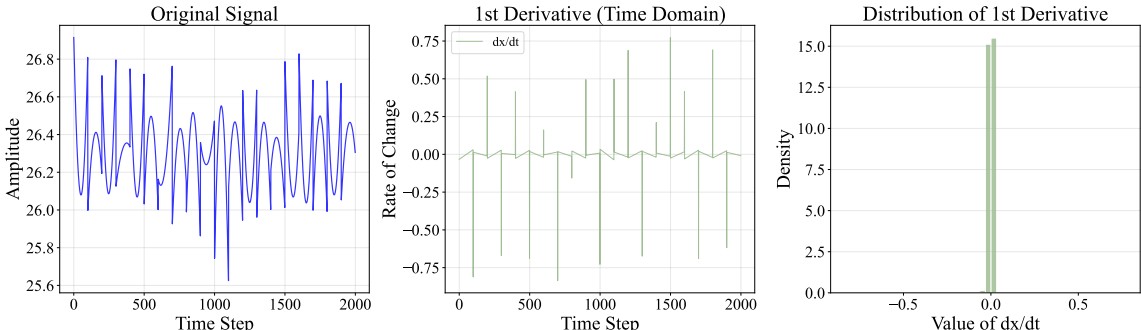

*(c)* CHESS generates smooth, continuous trajectories with derivatives highly concentrated around zero, demonstrating that our polynomial parameterization effectively eliminates non-physical noise and preserves temporal regularity.

*Figure 6.* Assessment of trajectory smoothness on the MeR dataset via first-order derivatives.

transitions and prevent underfitting.

For hyperparameter rank $R$, we determine the low-rank constraint $R$ using a *prior-guided strategy* based on the physical topology of the sensor array. For datasets with high-density sensor layouts, we observe significant inter-channel correlation. A sample in MeR dataset, as illustrated in Fig. 8, the signals across 30 distinct channels exhibit highly synchronized waveform profiles, implying a very low effective DoF. Guided by this prior, we aggressively set $R = 5$, which is sufficient to reconstruct the shared global dynamics while effectively filtering out channel-specific noise. Similar to this, for distributed setups like UCI-HAR, where sensors are mounted on three distinct anatomical locations (waist, chest, and ankle), the signals inherently group into three independent kinematic clusters. Accordingly, we align the rank with the physical constraints by setting $R = 3$. This ensures that the model preserves the distinct motion characteristics of each body part without enforcing artificial coupling between anatomically independent sensors.

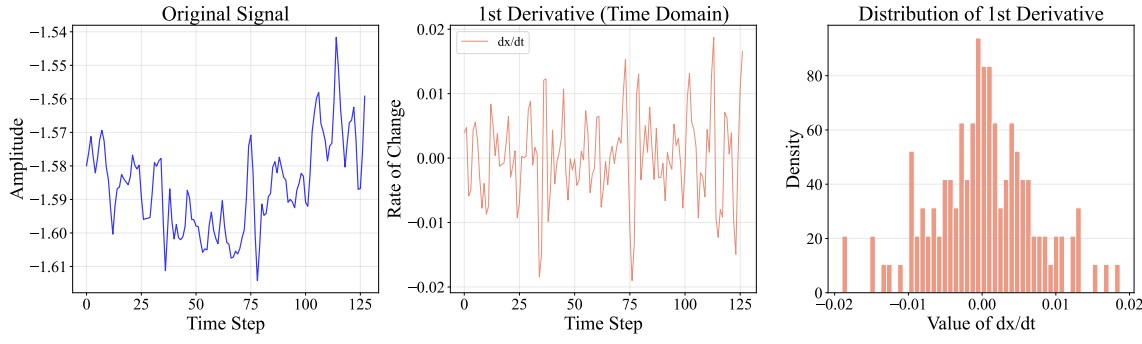

*(a)* Original signal in UCI-HAR dataset. The original sensor signal exhibits natural physical continuity.

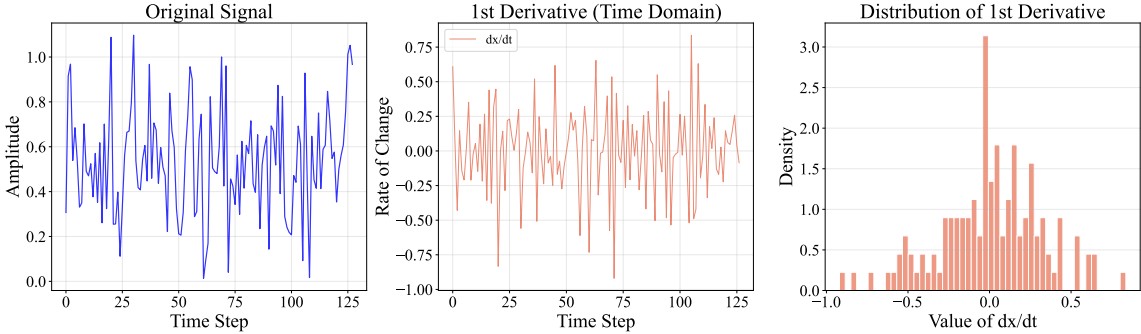

*(b)* The baseline NCFM, relying on discrete pixel-wise optimization, produces erratic fluctuations with a wide, heavy-tailed derivative distribution (range $\approx \pm 1$)

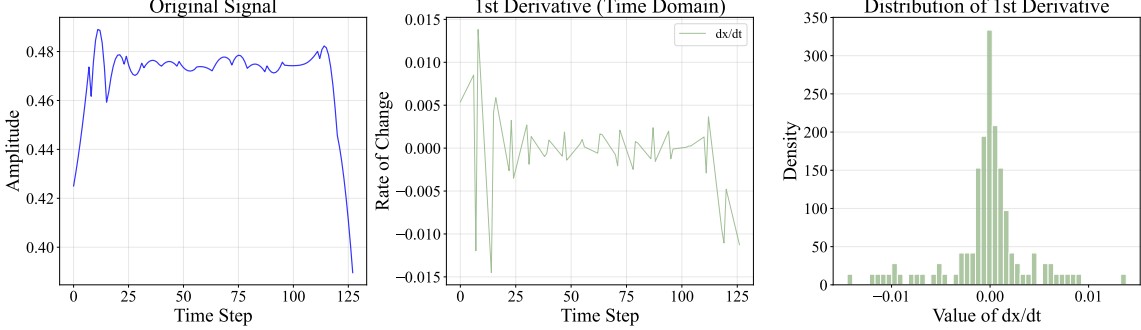

*(c)* CHESS generates smooth, continuous trajectories with derivatives highly concentrated around (range $\approx \pm 0.01$), demonstrating that our polynomial parameterization effectively eliminates non-physical noise and preserves temporal regularity.

*Figure 7.* Assessment of trajectory smoothness on the UCI-HAR dataset via first-order derivatives.

### E.5. Connection to Implicit Neural Representations

Our framework shares a conceptual lineage with Implicit Neural Representations (INRs) (Sitzmann et al., 2020; Shin et al.), which parameterize signals as continuous functions (e.g., neural fields) rather than discrete grids. However, unlike standard INRs that rely on black-box Multi-Layer Perceptrons (MLPs), CHESS adopts a structured, *interpretable basis expansion*. This distinction is crucial for scientific domains: MLPs are prone to spectral leakage (Tancik et al., 2020; Rahaman et al., 2019) and require iterative inference, our polynomial parameterization offers a closed-form analytical decoding with theoretically guaranteed approximation bounds (Trefethen, 2019). CHESS thus represents a step towards *Function First* DD, bridging the gap between classical signal processing theory and modern data condensation paradigms.

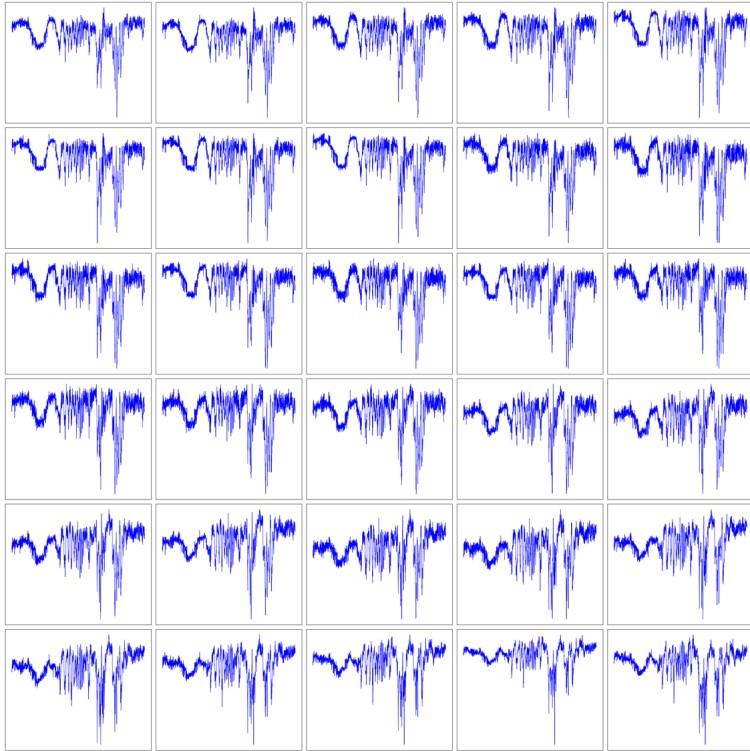

*Figure 8.* A sample signal in MeR dataset, we observe significant inter-channel correlation.

### E.6. Storage-Computation Trade-off

While CHESS achieves extreme storage compression by storing polynomial coefficients instead of raw signals, it necessitates a real-time decoding step during the training of downstream models. One might concern that this 'on-the-fly' synthesis introduces computational overhead. However, thanks to the recursive properties of Chebyshev polynomials, the evaluation cost is negligible compared to the backpropagation pass of modern deep networks (Defferrard et al., 2016).

## F. More Visualization Results

### F.1. More T-SNE

To intuitively evaluate the quality of the distilled data, we visualize the feature distributions of the original dataset and the synthetic samples generated by CHESS using t-SNE (Maaten & Hinton, 2008), as shown in Fig. 9. We extend this analysis across five diverse datasets: ActR, FacT, NTU-Fi, UCI-HAR, and PAMAP2. In the plots, grey dots represent the real data distribution, while red stars indicate our distilled samples. We provide visualizations for two condensation settings: 1 sample per class (SPC=1, top row) and 5 samples per class (SPC=5, bottom row). As observed, even with extreme condensation (SPC=1), the distilled samples generated by CHESS successfully locate themselves near the cluster centers of the real data, effectively capturing the representative prototypes of each class. When increasing the budget to SPC=5, the synthetic samples exhibit greater diversity and a broader coverage of the underlying data distribution. This alignment in the feature space demonstrates that CHESS preserves the essential discriminative information required for effective downstream classification.

### F.2. More Samples

Figures 10 – 14 show the synthetic samples generated by CHESS across different datasets. We can make two main observations from these plots. First, CHESS effectively preserves the inter-channel correlations. As shown by the synchronized changes across different sensor axes, our method captures the relationship between channels rather than treating each one separately. Second, the generated signals show high temporal smoothness. Unlike point-wise optimization

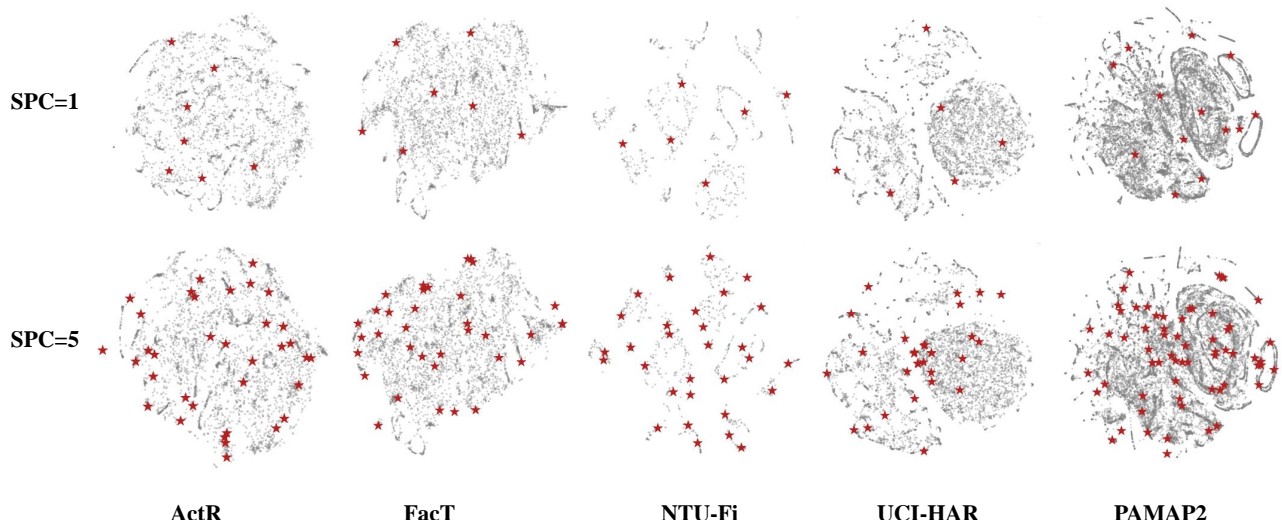

*Figure 9.* Feature distributions of the ActR, FacT, NTU-Fi, UCI-HAR and PAMAP2 dataset. Grey dots represent the original data, while red stars indicate distilled samples generated by CHESS. Top: SPC=1; Down: SPC=5.

methods that often produce noisy or jagged results, our approach ensures that the distilled patterns are smooth and continuous, matching the real characteristics of CTDS data.

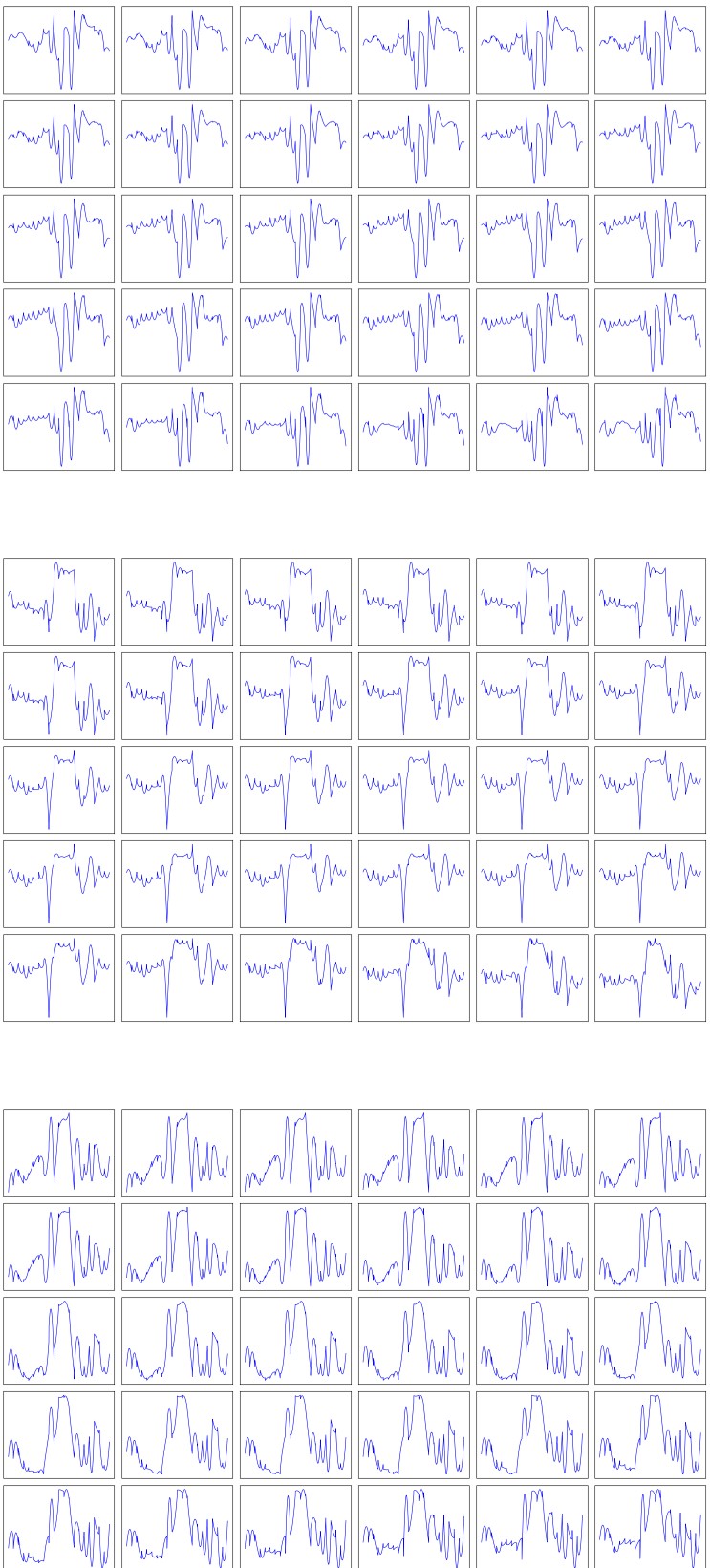

*Figure 10.* Visualizations of 3 representative distilled samples generated by CHESS on the ActR dataset.

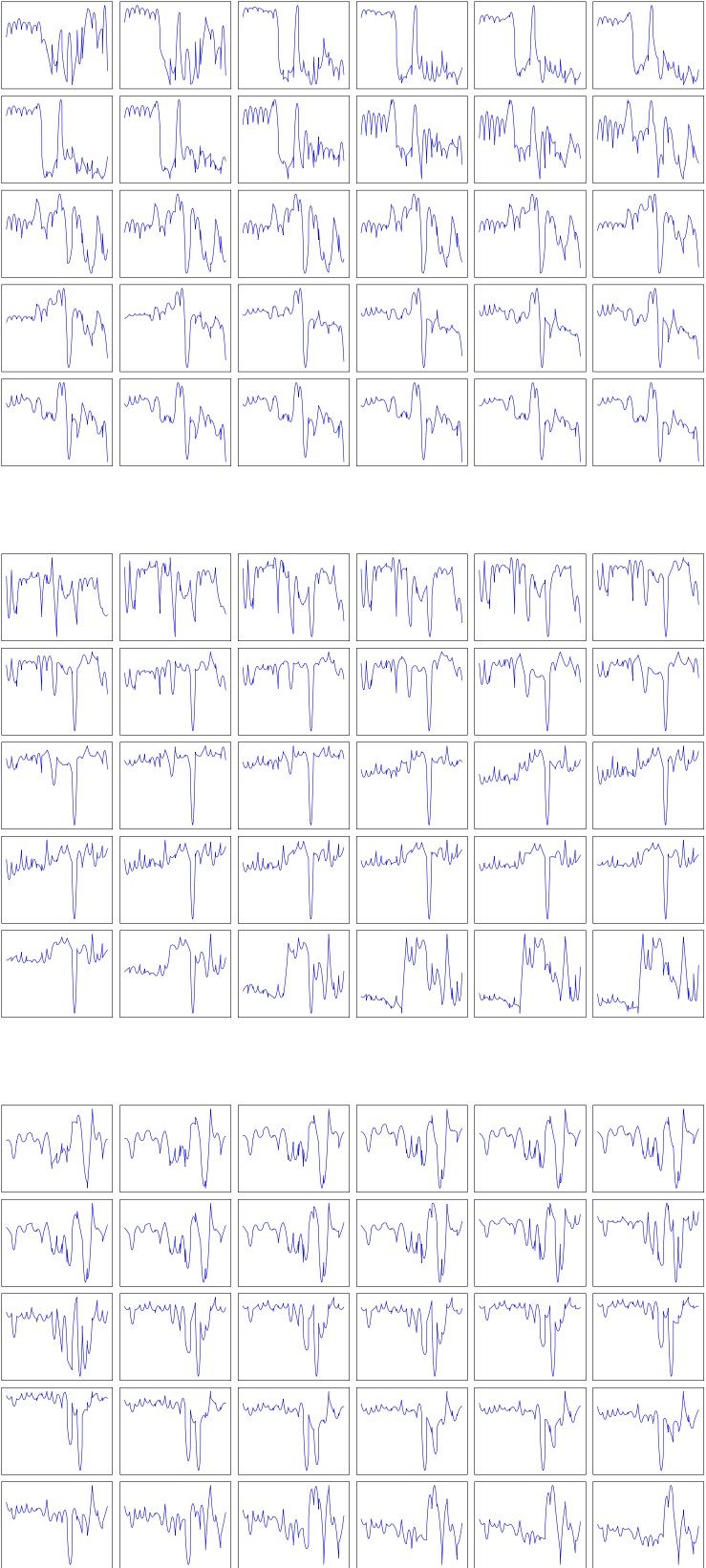

*Figure 11.* Visualizations of 3 representative distilled samples generated by CHESS on the MeR dataset.

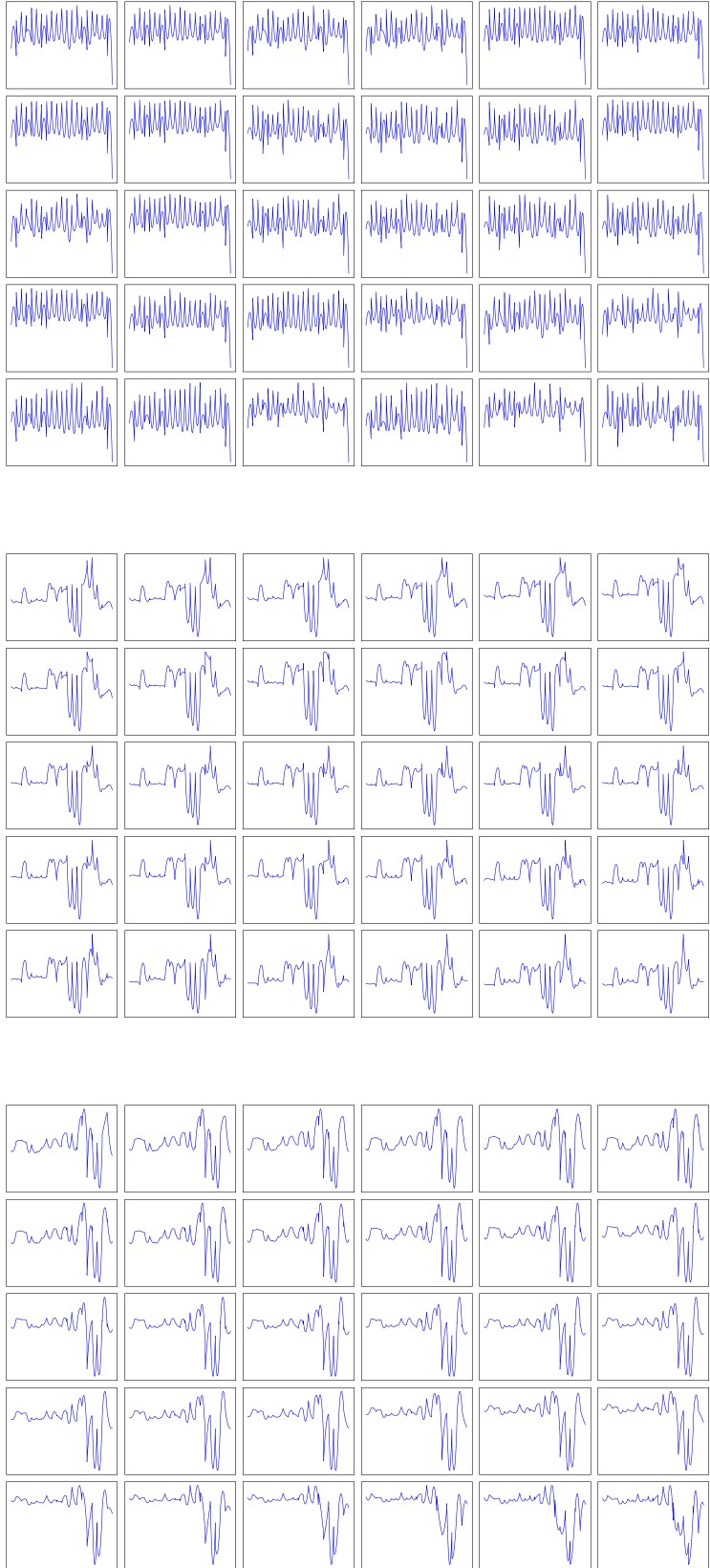

*Figure 12.* Visualizations of 3 representative distilled samples generated by CHESS on the FacT dataset.

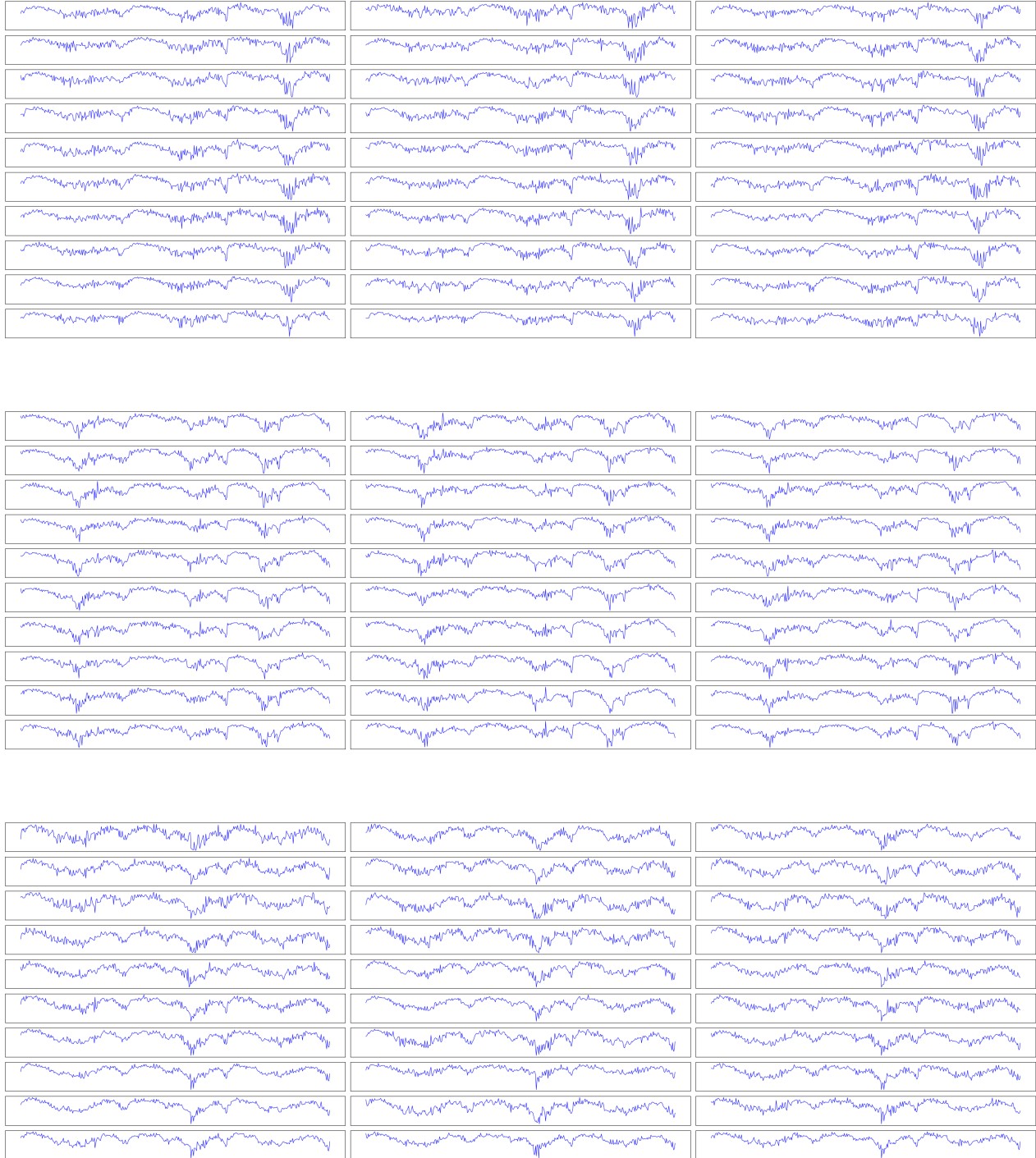

*Figure 13.* Visualizations of 3 representative distilled samples generated by CHESS on the NTU-Fi dataset. We plot first 30 channels.

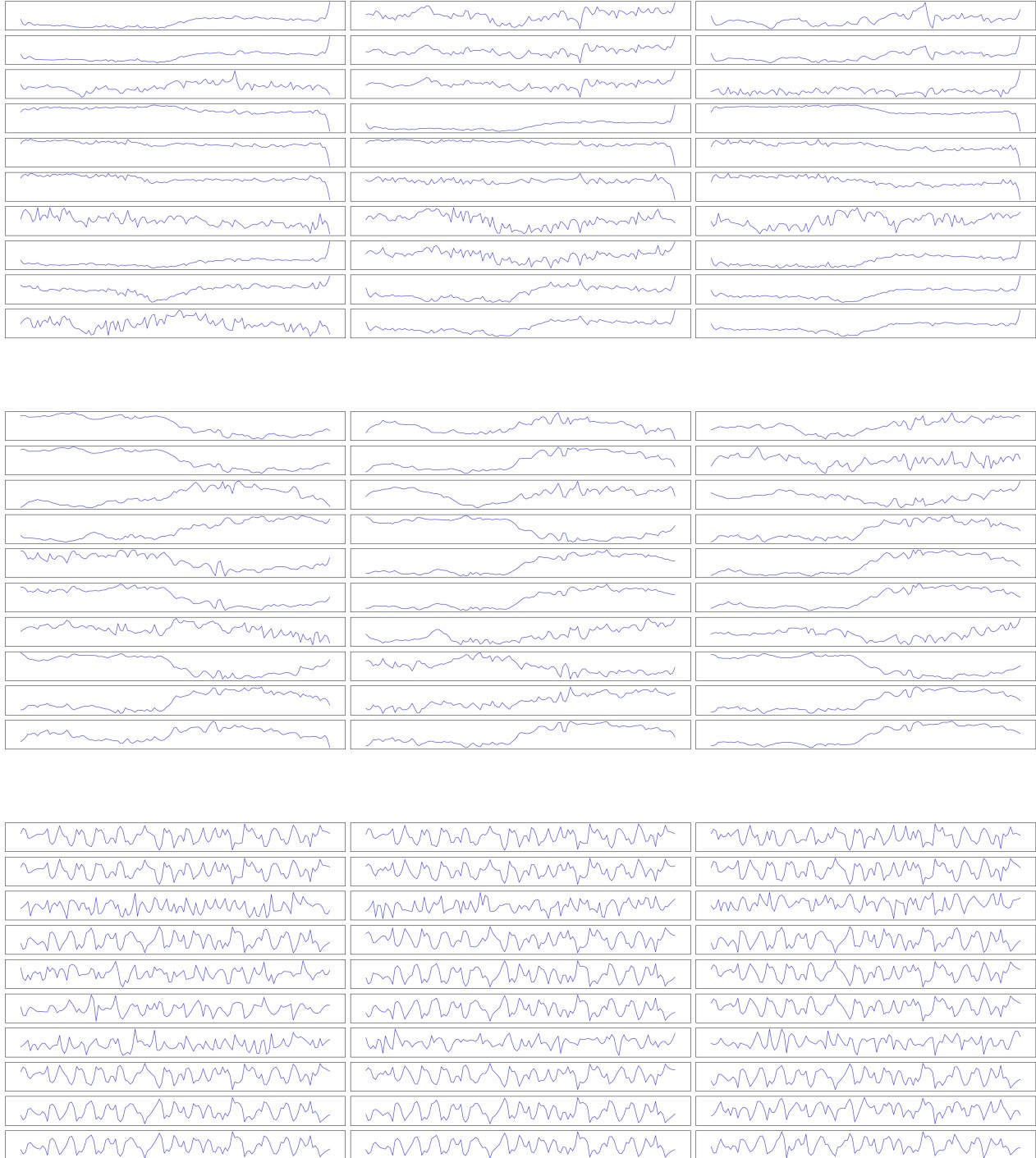

*Figure 14.* Visualizations of 3 representative distilled samples generated by CHESS on the PAMAP2 dataset. We plot first 30 channels.

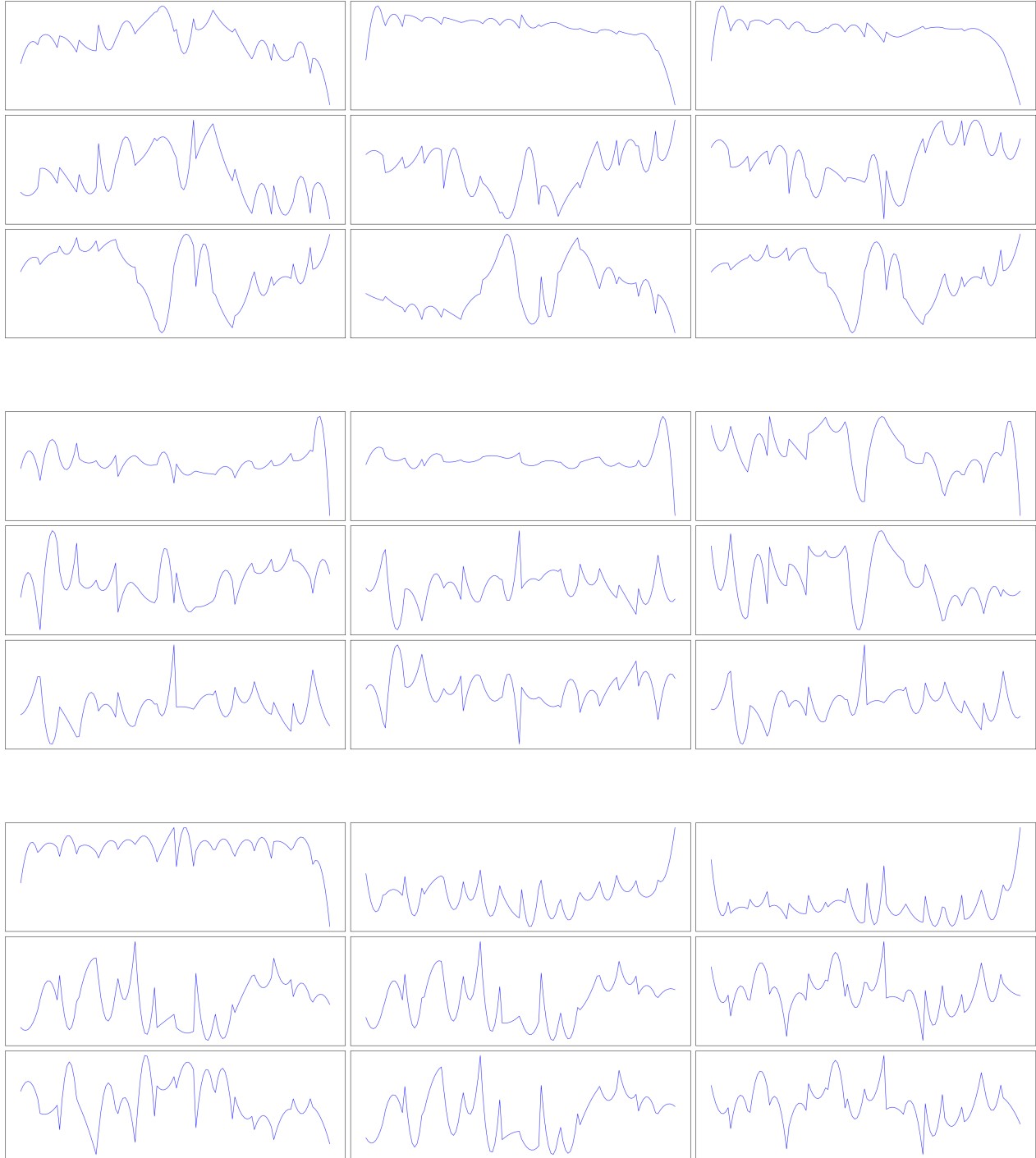

*Figure 15.* Visualizations of 3 representative distilled samples generated by CHESS on the UCI-HAR dataset.

