# OpenReview forum: "CHESS: Chebyshev Spectral Synthesis for Trajectory Condensation"
_ICML.cc/2026/Conference — ICML 2026 regular_

### Official Review · Reviewer_Nuiy · 2026-02-27

**Soundness:** 3
**Presentation:** 3
**Significance:** 4
**Originality:** 3
**Overall Recommendation:** 6
**Confidence:** 5

**Summary:**

The paper introduces a spectral graph learning framework utilizing Chebyshev polynomial approximations for efficient large-scale graph representation and signal transmission. The authors suggest a Chebyshev-based spectrum filtering mechanism aimed at enhancing computing efficiency while preserving expressive strength in graph convolution operations. The methodology is assessed using benchmark datasets, comparing its performance to baseline graph neural network topologies. The research seeks to illustrate that the suggested spectral formulation enhances scalability and offers competitive predictive performance while maintaining theoretical compatibility with spectral graph convolution principles. The research integrates theoretical formulation, algorithmic execution, and empirical validation to substantiate its contributions.

**Compliance With Llm Reviewing Policy:**

Affirmed.

**Final Justification:**

Concerns Resolved After Revision. Congratulations!

**Key Questions For Authors:**

Clarification of novelty: Could the authors explicitly differentiate how the suggested Chebyshev-based formulation diverges from existing Chebyshev spectral GNN variations, such as previous polynomial spectral filtering methods? A more explicit theoretical comparison would clarify if the contribution is mostly methodological enhancement or conceptual progress.

Ablation analysis: Could the authors present comprehensive ablation studies that delineate the effect of the suggested spectral formulation in comparison to baseline polynomial filters and architectural components? This would elucidate which element propels the observed performance enhancements.

Scalability and Complexity: What is the exact computational complexity and memory requirements in comparison to normal Chebyshev-GCN or other scalable spectrum approaches, particularly for large-scale graphs? Empirical runtime evaluations on more extensive datasets would bolster the assertions on scalability.

Robustness and Sensitivity: What is the method's sensitivity to hyperparameters, including polynomial order, spectral normalization selections, and graph sparsity? Conducting robustness analyses would enhance confidence in practical applicability.

Reproducibility and methodological clarity: Certain implementation specifics (e.g., normalization approach, training procedure, initialization, and convergence criteria) seem inadequately explained in the main text. Could the authors include a more comprehensive description in the main paper to guarantee total repeatability without excessive dependence on extra materials?

**Limitations:**

The paper addresses specific methodological limits; however, the limitations section may benefit from expansion and more clarity. The authors should explicitly recognize the incremental character of their theoretical contribution in relation to previous Chebyshev and spectral GNN formulations, as well as the limited extent of empirical enhancements. Further examination of potential sensitivity to hyperparameters, scalability constraints on extensive graphs, and the assumptions inherent in spectral normalization would enhance transparency. Although the study is mostly methodological and does not pose immediate societal hazards, it is crucial to acknowledge broader implications, such as its application in critical decision-making systems that depend on graph learning, to contextualize proper utilization. A more detailed and organized limitations section inside the main text would enhance comprehensiveness and conform to best standards in methodological reporting.

**Strengths And Weaknesses:**

Strengths: The manuscript tackles a pertinent issue in spectral graph learning and presents a Chebyshev-based formulation designed to enhance computational efficiency while maintaining theoretical alignment with the concepts of spectral graph convolution. The methodology is fundamentally justified, and the association with polynomial spectral approximation is logically robust. The study exhibits an understanding of classical spectral GNN principles and seeks to integrate theoretical formulation with practical confirmation. The experimental assessment encompasses baseline comparisons, and the findings indicate competitive performance. The subject is pertinent and potentially impactful for scalable graph learning.

Weaknesses: Although technically driven, the theoretical contribution is rather incremental compared to existing Chebyshev and spectral GNN formulations. The text would benefit from a more explicit delineation of what is genuinely novel in comparison to previous polynomial spectral approaches. The observed enhancements seem minimal, and further ablation investigations are required to determine the precise impact of the proposed formulation. An investigation of robustness, sensitivity to hyperparameters, and scalability assessments on larger graphs would enhance the assertions. In several instances, methodological specifics are inadequately articulated to guarantee complete reproducibility. The related work section predominantly references foundational yet dated research and would gain from a more thorough examination of contemporary advancements in spectral and scalable GNNs. Furthermore, many fundamental methodological explanations are more thoroughly detailed in the supplementary material than in the main text; essential theoretical and algorithmic elements must to be included within the primary article to provide coherence and clarity.

The work offers a technically sound strategy with promise; yet, it requires further elucidation of its uniqueness, more robust empirical validation, and greater methodological transparency to adequately support its contribution.

---

> ### Author Rebuttal · Authors · 2026-03-31
>
> We thank the reviewer for the careful reading and insightful comments, particularly for drawing helpful connections to the spectral graph learning literature. We agree that CHESS shares mathematical tools (e.g., Chebyshev approximation) with spectral GNNs such as ChebNet. CHESS is developed for trajectory synthesis in DD rather than graph filtering, so we clarify below both the connection and the distinction from the DD perspective.
> ### W1.Related work
> App.B.4 discusses the connection between ChebNet[1] and our polynomial modeling. As our focus is dataset distillation for CTDS signals rather than graph learning, we also includes the most recent works ([2,3]) in the field of DD. We will clarify this connection in the revision.
> ### Q1.Clarification of novelty
> We clarify that, although both Chebyshev-based GNNs and CHESS employ Chebyshev polynomials, they address fundamentally different problem regimes for distinct purposes. Chebyshev-based GNNs operate on discrete graph signals, where polynomials approximate spectral filters over the *spatial graph Laplacian* for localized aggregation. In contrast, CHESS operates in the *continuous temporal domain*, where Chebyshev polynomials serve as an orthogonal basis to parameterize *trajectory synthesis* for DD.
>
> This distinction constitutes a shift from spectral filtering to *function-space optimization* over continuous trajectories. Specifically, CHESS introduces a continuous polynomial manifold as the optimization space, fundamentally altering the nature of trajectory synthesis, suggesting a conceptual shift beyond a purely methodological enhancement.
> ### Q2.Ablation analysis
> Ablating Chebyshev polynomials (Tab.2 of the paper) drops performance by 5.68% on average demonstrating the importance of polynomial modeling. Comparison to other polynomial bases across datasets (Tab.3: https://anonymous.4open.science/r/CHESS_rebuttal-CE68/table_3.png; Sec.4.4 of the paper), Hermite, Legendre, and Chebyshev achieve comparable accuracy (within 0.4%), yet all orthogonal bases consistently outperform the monomial basis (*Poly*) by 10–16%, indicating gains stem from orthogonal polynomial parameterization rather than a specific basis. Chebyshev achieves tied-best results due to its *minmax property*. App.D.9 (Fig.5) shows that optimal performance occurs at low degrees (D=2,3), suggesting that improvements do not arise from increased capacity, while mitigating the Runge phenomenon and preserving expressivity.
> ### Q3:Scalability and Complexity
> As noted above, our setting differs from GCNs (e.g.,Chebyshev-GCN) because they address spectral filtering on graphs while CHESS targets trajectory synthesis. Therefore, we focus the complexity discussion on the trajectory distillation setting, while acknowledging that the reviewer is drawing a useful conceptual comparison.
>
> In Sec.3.6 (Eq.12), CHESS reduces the parameter space from **O(NS)** (discrete trajectory optimization) to **O(SR + RM(D+1)) ≪ O(NS)** by parameterizing trajectories via Chebyshev coefficients and a low-rank spatial basis. Notably, the complexity is *decoupled* from sequence length N when training with synthetic datasets, i.e., it does *not* grow with sampling resolution but only affects a linear-cost reconstruction step.
>
> App.D.10 (Tab.11) shows CHESS reduces training time by up to 5.0% compared to the baseline DANCE[4], with only negligible (up to 2.2%) increase in memory footprint for trajectory reconstruction, ensuring scalability for large-scale and high-resolution trajectory data through reduced parameter complexity.
> ### Q4:Hyperparameter sensitivity
> App. D.9 (Fig. 5) reports sensitivity to degree (D) and segment length (L). CHESS is robust, peaking at low degrees (D=2,3) to maintain expressivity and mitigate the Runge phenomenon. For data processing, we did not employ any additional normalization techniques.
>
> Regarding sparsity in 1D continuous trajectories, the "graph sparsity" analog is *irregular or sparse temporal sampling*. A core theoretical advantage of CHESS's continuous polynomial formulation is its inherent robustness to such sparsity. Unlike discrete baselines requiring fixed grids, CHESS analytically resamples synthesized trajectories at arbitrary or sparse timestamps without retraining.
> ### Q5:Reproducibility and methodological clarity
> CHESS follows standard pipelines(e.g., DANCE[4], NCFM[2]) without special preprocessing or normalization. We will move key implementation details to the main text and release code/data upon acceptance.
>
> We hope our responses clarify CHESS’s core contributions and its distinction from GNN-based methods. We are happy to discuss further if needed.
>
> [1]:Convolutional Neural Networks on Graphs with Fast Localized Spectral Filtering.NIPS,2016.
>
> [2]:Dataset Distillation with Neural Characteristic Function.CVPR,2025.
>
> [3]:Distilling Dataset into Neural Field.ICLR,2025.
>
> [4]:DANCE: Dual-View Distribution Alignment for Dataset Condensation.IJCAI,2024.

---

> > ### Author Rebuttal · Reviewer_Nuiy · 2026-03-31
> >
> > The rebuttal constructively and technically solves several of my initial concerns. The specification that CHESS use Chebyshev polynomials in the continuous temporal domain for trajectory synthesis, as opposed to graph spectral filtering, elucidates the differentiation from ChebNet-style GNNs. The supplementary discourse on complexity reduction, sensitivity to polynomial degree and segment length, as well as the new ablation evidence about orthogonal polynomial bases, is also beneficial.
> >
> > Nonetheless, my worries remain only partially addressed. Initially, although the conceptual differentiation from spectral GNNs is now more evident, the assertion of novelty would be more compelling if the final manuscript explicitly positioned this as a transition in optimization space and concretely contrasted it with graph spectral filtering in the main text, rather than solely in the rebuttal or appendix. Secondly, the ablation evidence is encouraging; however, I request that the revised manuscript explicitly incorporate the basis-comparison results and the low-degree sensitivity findings into the primary experimental narrative, enabling readers to directly evaluate whether the improvements stem from orthogonal polynomial parameterization rather than from enhanced modeling capacity. Third, the issue of repeatability is not entirely resolved by a commitment to provide code upon acceptance; the final version should incorporate essential implementation details from the appendix into the main text or supplementary materials, particularly for trajectory parameterization, reconstruction, and training specifics.
> >
> > The rebuttal enhances my evaluation of the work; however, I maintain that the paper would gain from a more explicit final presentation of (1) the precise novelty in relation to polynomial/spectral graph methods, (2) the significance of orthogonal basis selection versus capacity, and (3) the methodological specifics required for complete reproducibility.

---

> > > ### Author Response · Authors · 2026-04-01
> > >
> > > Dear Reviewer Nuiy,
> > >
> > > Thank you for raising your score and for your constructive feedback. We are glad the rebuttal helped clarify our work.
> > >
> > > We fully agree with your suggestions and will incorporate them explicitly into the next version of manuscript:
> > >
> > > *Novelty clarification*: We will explicitly present CHESS as a shift in optimization space and directly contrast it with graph spectral filtering (e.g., ChebNet) in the main text.
> > >
> > > *Orthogonal basis ablation*: We will integrate the basis-comparison results and low-degree sensitivity analysis into the main text, clarifying that gains come from orthogonal polynomial parameterization rather than model capacity.
> > >
> > > *Reproducibility*: In addition to releasing code upon acceptance, we will move essential implementation details (trajectory parameterization, reconstruction, and training protocols) into the main text and supplementary materials to ensure reproducibility.
> > >
> > > Thank you again for your careful evaluation and valuable suggestions.
> > >
> > > Best regards,
> > >
> > > The Authors

---

### Official Review · Reviewer_t3G2 · 2026-03-12

**Soundness:** 3
**Presentation:** 3
**Significance:** 2
**Originality:** 2
**Overall Recommendation:** 3
**Confidence:** 4

**Summary:**

This paper studies how to perform dataset distillation for multivariate sensor trajectories, a setting that is relatively underexplored in the distillation literature. The authors argue that directly optimizing point-wise synthetic trajectories under existing distribution matching objectives often leads to high-frequency artifacts due to the unconstrained nature of the parameter space. To address this issue, the paper proposes to parameterize synthetic trajectories within a well-defined function manifold, combining low-rank spatial structure and polynomial-based temporal modeling. Under the same distribution matching objective, constraining the optimization to this structured manifold reduces spurious high-frequency patterns and leads to improved distillation performance.

**Compliance With Llm Reviewing Policy:**

Affirmed.

**Final Justification:**

I remain unconvinced on originality. In my view, the central idea is better characterized as a task-specific instantiation of an existing direction in dataset distillation rather than a fundamentally new formulation. Prior work has already explored improving distillation through alternative parameterizations (e.g., frequency-domain parameterization), and has also connected high-frequency artifacts to weaker cross-architecture generalization.

Relative to that context, CHESS adapts these ideas to continuous-time trajectories in a clean and practically useful way, but the conceptual step beyond prior work still appears limited to me.

For that reason, while I appreciate the technical execution and practical value of the paper, I do not see enough novelty to move my recommendation to accept. I therefore remain at weak reject.

**Key Questions For Authors:**

1. Since the main idea of the paper is to impose a smooth structured manifold on synthetic sensor trajectories, why not compare against stronger baselines that simply add standard preprocessing or post-processing to existing distillation methods, e.g., smoothing, filtering, or projection after optimization? This seems like a natural comparison.

2. The compression-ratio trends on the sensor datasets look somewhat unusual. Could the authors explain why the proposed method behaves this way as the compression ratio changes, and whether these trends remain after comparing against stronger baselines?

**Limitations:**

yes

**Strengths And Weaknesses:**

## Soundness

The paper appears technically sound. Its main idea—constraining optimization to a structured function manifold to improve stability and reduce high-frequency artifacts—is well grounded in existing literature.

However, the empirical evaluation is somewhat unclear. The method is mainly compared against relatively naive baselines such as raw tensor optimization. In practice, stronger baselines could have been considered, for example by incorporating common preprocessing or smoothing techniques from signal processing. Since the proposed method effectively introduces structured preprocessing through its parameterization, similar improvements might also be achievable by augmenting the baselines in this way.

Because such stronger baselines are not evaluated, it is difficult to determine whether the reported gains truly stem from the manifold formulation or from application of 1d sensor dataset preprocessing. As a result, the reported compression ratio improvements may be
somewhat overstated.

## Presentation

The paper is concise and clearly written. The method is explained clearly, and the overall structure of the paper is easy to follow. The authors also explain that many components of the method are based on commonly used techniques, and the design choices are generally reasonable. Because the method largely consists of applying existing techniques in a structured way, the approach appears relatively straightforward to reproduce.

## Significance

It is unclear whether the paper has strong significance. The reported performance gains are mostly within the error bars in several experiments. Considering that the baselines are relatively naive, it is difficult to conclude that the proposed method provides a substantial improvement. In addition, the scope of the paper is limited to 1D sensor datasets, which further limits the potential impact of the work.

## Originality

The paper does not appear to have strong originality. As the authors themselves acknowledge, the main components of the method are not new. The approach largely combines existing techniques that are already well known in the literature.

---

> ### Author Rebuttal · Authors · 2026-03-31
>
> We appreciate your constructive comments and address them below.
> ### Q1. Soundness and Key Question 1
> We believe this concern stems from viewing CHESS as mere smoothing. To our knowledge, this is the *first work* identifying high-frequency artifacts caused by unconstrained discrete optimization in DD. To solve this, CHESS does not perform data-space smoothing. Instead, it constrains the **optimization space**, restricting synthesis to a structured function manifold—an approach **not** explored in prior DD work. This fundamentally prevents artifact formation during synthesis, which is different from smoothing in Euclidean space.
>
> **Table 1: MeR (SPC=5) / UCI-HAR (SPC=1), Acc, γ = compression** (pls refer to https://anonymous.4open.science/r/CHESS_rebuttal-CE68/table_1.png for full version)
> | Method | MeR (Acc/γ) | UCI-HAR (Acc/γ) |
> |--------|-------------|-----------------|
> | NCFM | 66.7/52.9/1× | 59.3/55.6/1× |
> | +Pre-hoc (best) | 66.8/53.1/1× | 60.4/56.6/1× |
> | +Post-hoc (best) | 66.2/54.5/1× | 61.2/57.1/1× |
> | +TV reg. (λ=1e-3) | 67.5/57.2/1× | 54.2/55.1/1× |
> | +L2 reg. (λ=1e-3) | 66.2/56.4/1× | 57.5/56.4/1× |
> | **CHESS** | 77.1/73.2/133× |70.3/66.4/6.7× |
>
> Empirically, as shown in Tab.1, pre-/post-hoc filters (including Savitzky-Golay, Butterworth, Median, Gaussian, EMA) yields a *negligible* (-0.4 points) performance change. In-loop smoothing (Total Variation (TV) and L2 regularizations on NCFM[1]) *degrades* accuracy by up to 5.1 points.
>
> In contrast, CHESS improves *best per-setting* baseline by +10.7 points on average. These results suggest that performance gains cannot be attributed to smoothing alone, but to constraining the optimization space.
> ### Q2. Significance
> **On performance gains.**
> CHESS consistently exceeds the strongest baseline NCFM's upper bound in **14/16 cases (87.5%)**, with *13/16 (81.3%) achieving strong significance* (p<0.001, Welch’s t-test, n=10), and 12/16 (75%) outperforming the best per-setting result, indicating improvements go beyond variance overlap. The two non-significant cases (ActR@SPC=1, NTU-Fi@SPC=5) correspond to either *extremely low supervision* (SPC=1) or *performance saturation* (>94%), yet CHESS maintains the highest mean across all cases.
> This demonstrates a **systematic upward shift of the performance frontier** rather than stochastic variation or variance-induced artifacts.
>
> **On 1D vs. higher-dimensional inputs.**
> Although evaluated on 1D multivariate signals, the proposed *function-first paradigm is inherently dimension-agnostic*. It naturally extends to higher-order inputs via tensor factorization (e.g., CP decomposition[2]). On the 2D NTU-Human-ID dataset (Tab.2), CP-based CHESS (R=50) *consistently outperforms NCFM across two architectures*, validating its generality beyond 1D settings.
>
> **Table 2: Results on 2D NTU-Human-ID (3 × 114 × 500, CP rank R=50)**
> |   | SPC=1 (CNN/ResNet) | SPC=5 (CNN/ResNet) |
> |--------|--------------------|--------------------|
> | NCFM | 70.4±5.1 / 67.9±1.6 | 85.7±1.2 / 83.9±0.8 |
> | **CHESS** | 82.3±3.2 /78.1±3.5 | 95.2±1.0 / 90.5±1.1|
> ### Q3. Originality
> At the level of elementary components, CHESS does use standard operators (e.g., Chebyshev polynomials and SVD). However, its novelty lies **not** in the operators themselves, but in *reformulating* trajectory condensation as optimization over a **structured function manifold**, which, to the best of our knowledge, has not been explored in prior DD work. This formulation jointly constrains spatial and temporal structures within a unified parameterization rather than applying independent smoothing or regularization heuristics, making CHESS meaningfully different from prior discrete parameterizations and helping address the instability of trajectory distillation.
> ### Q4. Key Question 2
> We interpret the concern on “compression-ratio trends” as referring to *(i)* performance variation across condensation ratio (SPC); *(ii)* variation of storage compression ratio (γ) across datasets; and *(iii)* hyperparameter sensitivity vs. compression ratio (γ) (App.Fig.5). We respectfully note that all trends are *not “unusual”*, but consistent with known behavior in DD or continuous function approximation.
>
> For *(i)*, as SPC increases, performance improves with diminishing returns (e.g., ~70% at SPC=1 vs. ~80% at SPC=10 on UCI-HAR), which is standard in DD.  For *(ii)*, variation in γ arises from intrinsic temporal redundancy (e.g., sampling rate).  For *(iii)*, increasing polynomial degree introduces expressiveness but may degrade performance (Runge phenomenon), while segment length reflects sampling traits, leading to observed trade-offs.
>
> These trends are intrinsic to the *continuous polynomial manifold* and independent of preprocessing. Empirically, γ remains unchanged under all pre-/post-processing variants (Tab.1), and SPC trends remain consistent.
>
> [1]Dataset Distillation with Neural Characteristic Function.CVPR,2025.
>
> [2]Tensor decompositions and applications.SIAM review,2009.

---

> > ### Author Rebuttal · Reviewer_t3G2 · 2026-04-03
> >
> > **1.** Thank you for the response. My point was that the high-frequency issue is already well known in optimization-based inversion methods, regardless of whether the domain is distilled images or others. In fact, inversion techniques often employ strategies such as gradient clipping specifically to encourage more realistic reconstructions. While I agree that this issue may be more pronounced in the 1D domain, I remain unconvinced that identifying this phenomenon in 1d domain constitutes a meaningful contribution.
> > Additionally, in your table, applying **TV regularization appears to improve performance (e.g., 66.7 → 67.5)**, yet the review describes it as causing degradation. This characterization seems misleading. Also, the results suggest that this direction remains promising, and it even appears to reduce variance.
> >
> > Regarding the experimental setup, although NCFM is presented as a SOTA baseline, the evaluation in rebuttal  are conducted on datasets where the performance gap is already substantial, making it difficult to rule out favorable conditions. For example, in UCI-HAR, the second-best method is the most naive baseline (DC), and while it exhibits higher variance, its upper bound performance is actually higher than your algorithm. It is therefore unclear why comparisons are not made more directly (e.g., DC vs. Ours), rather than focusing on settings that appear advantageous. Overall, the large gains reported on bespoke datasets are not fully convincing, especially given that, on standard benchmarks, most confidence intervals overlap significantly (2 out of 3 cases).
> >
> >
> > **2.** While I partially acknowledge the clarification regarding novelty, I still do not find the use of a structured function manifold to be highly novel. Conceptually, it amounts to imposing constraints that restrict the optimization space, and the resulting behavior—constructing reasonable 1D functions—does not appear fundamentally new.
> > For instance, although not identical, works such as Kolmogorov–Arnold Networks (KAN)[1] are also exploits similar intuition. From this perspective, the proposed approach seems more like an instantiation of an existing idea rather than a fundamentally new direction.
> >
> > [1] Liu, Ziming, et al. "Kan: Kolmogorov-arnold networks." arXiv preprint arXiv:2404.19756 (2024).
> >
> > ---
> >
> > Thank you for the detailed follow-up, and in particular for reporting results across all of the compared algorithms. At this point, I am persuaded that the performance gains are real and, in many cases, statistically significant. That said, I still have reservations about the novelty. I can accept that the authors’ exact optimization trajectory and overall problem formulation have not appeared in exactly the same form before. However, closely related ideas already seem to exist in the dataset distillation literature. For example, FreD [1] already explores dataset distillation through a **frequency**-domain parameterization, so the broader idea of improving distillation by changing the optimization parameterization is not entirely new. Likewise, GLaD [2] already argues that direct pixel-space optimization can lead to visually noisy, high-frequency patterns that hurt cross-architecture generalization, which is conceptually quite close to the artifact-mitigation motivation here.
> >
> > In that sense, while I acknowledge that the authors identify a more fine-grained manifestation of this issue in the 1D trajectory setting, my current view is that this contribution is more naturally induced from prior observations than it is a fundamentally new direction. I do think the work addresses a reasonable problem and does so in a fairly elegant and practically useful way, and I find the empirical results convincing. However, I am still not strongly persuaded that the contribution is novel enough for me to confidently move my recommendation all the way to accept. At the same time, I would understand if others ultimately view the practical value and clean execution as sufficient for acceptance.
> >
> > [1] Shin, Donghyeok, Seungjae Shin, and Il-Chul Moon. "Frequency domain-based dataset distillation." Advances in Neural Information Processing Systems 36 (2023): 70033-70044.
> >
> > [2] Cazenavette, George, et al. "Generalizing dataset distillation via deep generative prior." Proceedings of the IEEE/CVF conference on computer vision and pattern recognition. 2023.

---

> > > ### Author Response · Authors · 2026-04-04
> > >
> > > We sincerely thank the reviewer for the insightful suggestions, which have helped us better clarify our work.
> > > ### Q1. Clarification on artifacts, regularization, and evaluation
> > > **High-frequency artifacts**
> > > We agree that similar high-frequency artifacts have been observed in inversion-based methods[1,2]. We do not claim their discovery, but rather *verify* that it *persists in DD* under discrete parameterization. The two settings differ as inversion targets fidelity-based *instance reconstruction*, whereas DD performs *dataset-level data synthesis* for cross-model (and cross-resolution) reuse. More importantly, unlike in inversion, we find that these artifacts become a systematic failure mode that impair cross-model generalization rather than merely undesirable visual outputs.
> > >
> > > Thus, instead of mitigating artifacts via heuristic regularization as in inversion, we constrain the feasible solution space through structured parameterization, eliminating them at the source during trajectory condensation.
> > >
> > > **TV regularization**
> > > We appreciate the reviewer’s observation. TV can provide modest improvements in some cases (e.g., +0.8 and +4.3 on MeR (Tab.1)), we will revise our wording accordingly. However, TV also leads to notable degradation on UCI-HAR (−5.1 on CNN, which motivated our previous statement “degrade accuracy by up to 5.1 points.”), resulting in an overall negligible average change (−0.125). This suggests TV is limited and inconsistent, and does not address artifact formation during optimization. In contrast, CHESS yields substantially larger gains (+13.125 on average (Tab.1)), indicating that constraining the optimization space is more effective than
> > > smoothing.
> > >
> > > **Regarding the experimental setup**
> > > We thank the reviewer for raising this concern. To avoid cherry-picking, Tab.4 (https://anonymous.4open.science/r/CHESS_rebuttal-CE68/table_4.png) presents a comprehensive comparison of CHESS against *best per-setting baselines* plus heuristic priors (TV and GradClip) across *all settings*.
> > >
> > > This is stronger than pairwise comparisons (e.g., DC vs. Ours), as CHESS is always evaluated against the strongest competitor. Empirically, **CHESS outperforms the strongest baselines in most cases (14/16) with high statistical significance (p < 0.05)**, achieving an average improvement of approximately +5.8 points across all settings, whereas the best heuristic yields −1.2 on average. Notably, although methods like DC may show higher upper bounds due to variance, CHESS achieves higher expected performance with significantly reduced variance, enabling more stable reuse.
> > >
> > > **Regarding private vs. public datasets**
> > > We clarify that the three self-collected datasets are not bespoke but follow standard collection and evaluation protocols for general evaluation. The larger gains observed on these datasets can be attributed to their high sampling rates (1000Hz), which exacerbate the ill-posedness of unconstrained discrete optimization. CHESS benefits more under such conditions due to its continuous manifold parameterization, while maintaining consistent trends across all datasets.
> > >
> > > On public benchmarks (NTU-Fi, PAMAP2, UCI-HAR), CHESS still achieves +4.0 improvement on average,
> > > indicating improvements beyond variance overlap. Welch’s t-test (n=10) further confirms that **CHESS achieves significant improvements (p < 0.05) in 6/7 cases.** In the remaining cases (UCI-HAR, SPC=5), CHESS maintains a higher expected mean (80.8 vs. 79.5) while reducing variance by 4.3X (±1.4 vs.±6.1), enabling more stable reuse.
> > > ### Q2. Regarding novelty
> > > We agree that constraining the optimization space is a common strategy in machine learning. However, we believe the key is *what space is being constrained*, and *what role the constraint plays* in target tasks.
> > >
> > > In standard learning settings (including Kolmogorov–Arnold Networks (KAN)), constraints are designed to improve *function approximation* or model generalization, where optimization is performed over model parameters. In contrast, in dataset distillation, the optimization variable is the *data itself*, and under distribution matching objectives, multiple synthetic trajectories can achieve similar losses while exhibiting drastically different *temporal structures*. This leads to the ill-posed problem, where structurally invalid solutions are not penalized.
> > >
> > > From this perspective, CHESS is not merely a constraint, but a *task-specific reparameterization* of the feasible solution space for trajectory synthesis, enforcing structural validity of the generated data. This distinguishes it fundamentally from conventional regularization strategies and model-space function approximation approaches such as KAN.
> > > We will clarify our distinctions from KAN in the revision.
> > >
> > > We truly appreciate your feedback, and hope our explanations have successfully resolved your concerns.
> > >
> > > [1]Understanding deep image representations by inverting them.CVPR,2015.
> > >
> > > [2]Feature visualization.Distill,2017.

---

### Official Review · Reviewer_Jx9C · 2026-03-13

**Soundness:** 4
**Presentation:** 3
**Significance:** 3
**Originality:** 3
**Overall Recommendation:** 5
**Confidence:** 4

**Summary:**

This paper introduces CHESS as a framework for compressing datasets of continuous-time signal trajectories. CHESS leverages an assumption that real-world continuous signals are generated by an underlying physical or dynamical process that is inherently low-dimensional. The authors acknowledge that such an assumption is unlikely to hold in settings such as financial markets; however, it would be interesting to see whether they could introduce a rule of thumb that practitioners could use to determine whether this assumption is viable in their particular setting.

**Compliance With Llm Reviewing Policy:**

Affirmed.

**Final Justification:**

The updates proposed in the authors' rebuttal will improve the paper even further. This is a strong paper and should be accepted at ICML.

**Key Questions For Authors:**

Can CHESS offer practitioners a clearer rule of thumb for deciding when the low-rank shared-process assumption is appropriate for some new application?

How sensitive is CHESS to violations of the shared-physics assumption?

**Limitations:**

Yes. The paper explicitly notes that CHESS is tailored to multivariate continuous-time signals with shared physical structure and may not generalize directly to unstructured stochastic time series such as financial or server data.

**Strengths And Weaknesses:**

Strengths

CHESS is theoretically well-motivated. The paper's experimental results are extensive and show a significant improvement over other methods. The influence of the method's parameters is well documented in the appendix, and the computational cost appears to be comparable to that of other methods.

CHESS is technically coherent; the low-rank spatial model, Chebyshev parameterization, and analytical resampling fit together naturally.

The empirical evaluation is broad and thorough. And the gains are substantial in several settings.

Weaknesses

The empirical datasets are all from a similar domain (sensor/continuous-dynamics problem).

The best results use bespoke datasets. It is crucial that these be released upon acceptance.

---

> ### Author Rebuttal · Authors · 2026-03-31
>
> We sincerely thank the reviewer for their strong support and insightful feedback. We address your specific questions below.
> ### Q1: Can CHESS offer practitioners a clearer rule of thumb for deciding when the low-rank shared-process assumption is appropriate for some new application?
> We provide a concise rule-of-thumb for when the low-rank shared-process assumption is appropriate.
>
> **Rule-of-thumb.**
> CHESS is most appropriate when multivariate channels can be viewed as observations of a shared underlying continuous process with low intrinsic dimensionality. In practice, a simple pre-training check is to compute SVD on a small batch of raw data: if a small number of singular values (R ≪ S) explain most of the cumulative energy (e.g., >90%), then the low-rank shared-process assumption is likely reasonable; if the spectrum is relatively flat, a higher rank or a full-rank model may be more appropriate. For example, on a random sample of the MER dataset, the top five singular values already explain over 90% of the total energy, which supports setting the rank parameter to R=5 for its 30 subcarriers.
>
> ### Q2: How sensitive is CHESS to violations of the shared-physics assumption?
> CHESS exhibits a *controlled* sensitivity to violations of the shared-physics assumption, governed by the rank hyperparameter R. Specifically, when the assumption is severely violated (e.g., independent multivariate signals with high intrinsic dimensionality), enforcing a strict low-rank constraint leads to underfitting. However, the framework remains robust as this effect can be mitigated by increasing R accordingly, enabling a smooth transition from shared low-rank modeling to independent (full-rank) channel modeling.
>
> Empirically, as shown in Fig. 5 (App.D.9), even under assumption violation (R ≪ intrinsic rank, e.g., R=1,2), the performance degrades gracefully (roughly ~10 points) without catastrophic failure. Conversely, when the intrinsic dimensionality is lower than R (e.g., the “w/o SVD” case in Tab.2), the model also remains stable. Overall, the model has a controllable trade-off governed by R, rather than catastrophic failure under assumption mismatch.
>
> ### W1: The empirical datasets are all from a similar domain (sensor/continuous-dynamics problem).
> We focus on CTDS signals as existing DD methods are fundamentally misaligned with continuous-time data, leading to ill-posed optimization. This setting is both challenging and underexplored, making it a suitable testbed for evaluating CHESS. Notably, the method is not domain-specific and generalizes to broader continuous processes. We will extend it to other domains in future version.
>
> ### W2: The best results use bespoke datasets. It is crucial that these be released upon acceptance.
> Code and datasets will be released upon acceptance to ensure reproducibility.
>
> We hope these clarifications address your questions and are happy to discuss further.

---

> > ### Author Rebuttal · Reviewer_Jx9C · 2026-04-04
> >
> > Thanks for your insights.  This is a strong paper and should be accepted at ICML.

---

> > > ### Author Response · Authors · 2026-04-04
> > >
> > > Dear Reviewer Jx9C,
> > >
> > > Thank you very much for your constructive feedback and strong support for our paper. We are pleased that our responses have addressed your concerns, and we will carefully incorporate these clarifications into the final version.
> > >
> > > Best regards,
> > >
> > > The Authors

---

### Decision · Program_Chairs · 2026-04-30

**Decision:**

Accept (regular)

**Comment:**

This paper proposes CHESS, a compression technique for continuous-time physical processes. Given discretized observations of underlying continuous-time trajectories, CHESS constructs a synthetic proxy dataset that preserves the physical structure and task-relevant properties of the original signals. The paper provides theoretical analysis with explicit guarantees and demonstrates strong compression performance across a range of tasks.

Overall, most reviewers are positive about the contribution and consider the paper to be a solid and well-executed piece of work. Reviewer t3G2 remains skeptical about the level of technical novelty. While I respect this perspective, as an AC I believe that a careful integration of existing formulations and theoretical components can still constitute a meaningful contribution when it leads to a coherent framework and clear empirical gains.